# $\partial^\infty$-GRID: A NEURAL DIFFERENTIAL EQUATION SOLVER WITH DIFFERENTIABLE FEATURE GRIDS

**Navami Kairanda[1], Shanthika Naik[2], Marc Habermann[1], Avinash Sharma[2], Christian Theobalt[1], Vladislav Golyanik[1]**
[1]Max Planck Institute for Informatics, Saarland Informatics Campus
[2]Indian Institute of Technology Jodhpur

## ABSTRACT

We present a novel differentiable grid-based representation for efficiently solving differential equations (DEs). Widely used architectures for neural solvers, such as sinusoidal neural networks, are coordinate-based MLPs that are both computationally intensive and slow to train. Although grid-based alternatives for implicit representations (*e.g.*, Instant-NGP and K-Planes) train faster by exploiting signal structure, their reliance on linear interpolation restricts their ability to compute higher-order derivatives, rendering them unsuitable for solving DEs. Our approach overcomes these limitations by combining the efficiency of feature grids with radial basis function interpolation, which is infinitely differentiable. To effectively capture high-frequency solutions and enable stable and faster computation of global gradients, we introduce a multi-resolution decomposition with co-located grids. Our proposed representation, $\partial^\infty$-*Grid*, is trained implicitly using the differential equations as loss functions, enabling accurate modelling of physical fields. We validate $\partial^\infty$-*Grid* on a variety of tasks, including the Poisson equation for image reconstruction, the Helmholtz equation for wave fields, and the Kirchhoff-Love boundary value problem for cloth simulation. Our results demonstrate a 5–20× speed-up over coordinate-based MLP-based methods, solving differential equations in seconds or minutes while maintaining comparable accuracy and compactness. See our project page: https://4dqv.mpi-inf.mpg.de/DInf-Grid/.

## 1 INTRODUCTION

Finding representations and parametrisations for fields, which are both accurate and fast to evaluate, is challenging and has many applications in many machine learning domains (vision, graphics, simulation of physical systems, solving PDEs, engineering) (Xie et al., 2022). We desire representations that model global structure and capture high-frequency as well as local details. Not only do we want to represent known signals well, but also to recover *unknown* fields by modelling physical systems governed by differentiable equations. At their core, many of these problems involve mapping spatial or spatio-temporal coordinates $\mathbf{x} \in \mathbb{R}^d$ to signals or fields $\mathbf{u}(\mathbf{x}) \in \mathbb{R}^m$. Such systems governed by differential equations can be represented as implicit functions $\mathcal{F}$ of the form:

$$\mathcal{F}\left(\mathbf{x}, \mathbf{u}, \nabla_{\mathbf{x}}\mathbf{u}, \nabla_{\mathbf{x}}^2\mathbf{u}, \ldots; g(\mathbf{x})\right) = 0, \tag{1}$$

where $g$ represents known vectors or functions controlling the system.

Replacing the traditional numerical solvers for equation 1 with neural networks has gained significant traction in recent years, particularly in the context of physics-informed machine learning (Karniadakis et al., 2021; Hao et al., 2022). The optimisation of network weights can emulate the process of solving physical equations; we refer to such networks as neural solvers for brevity. Current neural solvers are predominantly based on coordinate-based multi-layer perceptrons (MLPs) or extensions thereof for efficiency, accuracy, etc.; while effective, such solvers exhibit several limitations. For instance, they tend to favour low-frequency fitting of signal $\mathbf{u}$, requiring techniques like positional encoding (Vaswani et al., 2017) to handle high-frequency signals. Moreover, not all networks are suitable for solving DEs, as they are not differentiable beyond the first order with respect to the input coordinates (*e.g.*, those with ReLU activation lead to $\nabla_{\mathbf{x}}^2\mathbf{u} = 0$). Siren (Sitzmann et al., 2020), a

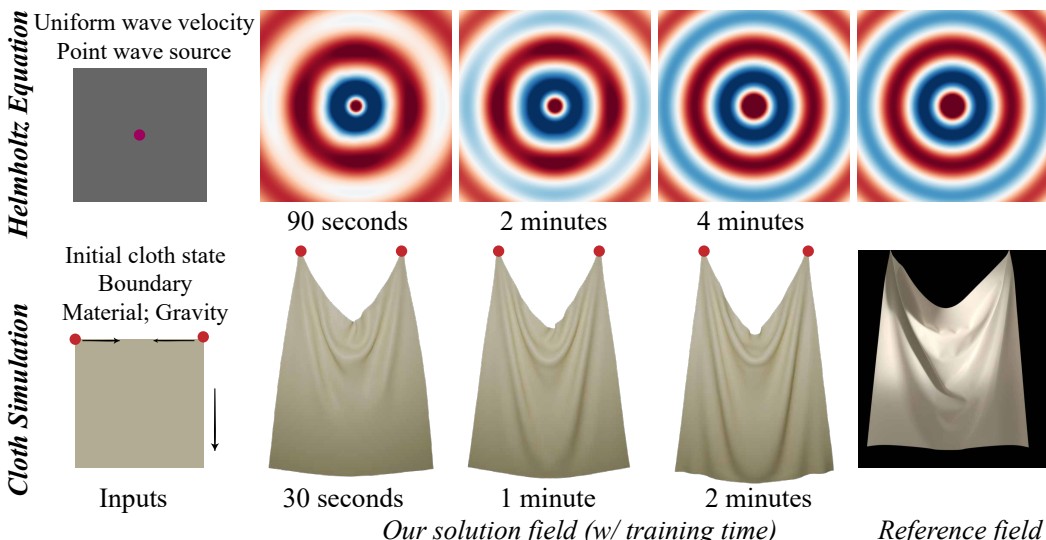

**Figure 1: Our proposed method, $\partial^\infty$-*Grid*, is a feature grid-based representation capable of accurately modelling signals and their higher-order derivatives.** We demonstrate its effectiveness in solving complex differential equations, including the Helmholtz equation (top) and neural cloth simulation (bottom) (Kairanda et al., 2024) (reference adapted from (Clyde et al., 2017)), achieving both high speed and accuracy.

sinusoidal activation-based architecture, addresses some of these issues by being infinitely differentiable and capable of fitting high-frequency signals, making it a preferred choice as a neural field for physical problems as in equation 1. However, Siren and other neural solvers do not account for the spatial structure of signals and their derivatives in their architecture. Consequently, they are computationally expensive, requiring long training times, sometimes up to several hours. On the other hand, to overcome the slow training and inference of neural fields (Mildenhall et al., 2020), explicit grid-based or hybrid architectures (Sara Fridovich-Keil and Alex Yu et al., 2022; Müller et al., 2022; Cao & Johnson, 2023) that leverage the spatial locality have been proposed for scene representation. While they offer significantly faster training, they rely on $d$-linear interpolation, which is not differentiable at the grid boundaries ($\nabla_{\mathbf{x}}\mathbf{u} = 0$) and not infinitely often differentiable in the grid interiors ($\nabla_{\mathbf{x}}^2\mathbf{u}, \cdots = 0$). This severely limits their applicability as neural representations for solving DEs.

To address the computational overhead of neural solvers and the failure of grid-based representations (Fridovich-Keil et al., 2023; Müller et al., 2022) in solving differential equations $\mathcal{F}(.) = 0$, we propose a novel feature grid representation capable of *accurately modeling fields* $\mathbf{u}$ *and their derivatives* $\nabla_{\mathbf{x}}\mathbf{u}, \nabla_{\mathbf{x}}^2\mathbf{u}, \ldots$. We combine the efficiency of feature grids with a differentiable interpolation technique supporting higher-order derivatives for effectively solving DEs. Our method leverages radial basis function (RBF) interpolation, which is smooth and infinitely differentiable, where the gradients can be computed analytically using automatic differentiation. To represent high-frequency details as well as the low-frequency global structure, we propose a multi-resolution co-located feature grid representation. Our approach offers several advantages. First, the feature grid localises learning, significantly speeding up training compared to global neural solvers. Second, our method provides continuous parameterisation despite using a grid, allowing queries at arbitrary points similar to neural fields. Consequently, it achieves accuracy comparable to (Sitzmann et al., 2020) or better than (Raissi et al., 2019), *i.e.*, MLP-based methods, while being significantly (up to $20\times$) faster. In summary, core technical contributions of our $\partial^\infty$-*Grid* method are as follows:

- A feature grid-based representation for physical fields and a formulation recovering fields from differential equations (Sec. 3.1).
- Differentiable RBF interpolation for accurately modelling higher-order derivatives (Sec. 3.2).
- Multi-resolution grids for faster global gradient flow (Sec. 3.3).

We validate $\partial^\infty$-*Grid* on diverse physical problems; see Fig. 1, where we learn the solutions for

- *Image with Poisson Equation*: A mapping from 2D coordinates to colours for high-resolution images using only gradient or Laplacian supervision;
- *Helmholtz Equation*: Mapping a spatial domain to complex steady-state wavefields;

- *Cloth Simulation*: Mapping 2D material space to 3D deformations for cloth quasi-statics, modelled as a thin-shell boundary-value problem;
- *Eikonal Equation*: A signed distance function learned directly from oriented point cloud;
- *Advection and Heat Equations*: Spatio-temporal transport and diffusion of physical quantities.

We prepared the source code for release, which can act as a drop-in model for both representing fields and solving fields from governing equations.

## 2 RELATED WORK

Traditionally, differential equations are solved with numerical methods (Butcher, 2016). Classical solvers discretise governing equations but struggle with data incorporation, meshing, and high-dimensional costs, motivating neural representations as a promising alternative.

**Neural Fields and DE Solvers.** Neural fields are continuous and non-linear mapping functions parametrised by trainable neural network weights. They have been widely used in representing radiance fields (Mildenhall et al., 2020), signed distance and occupancy fields (Park et al., 2019; Mescheder et al., 2019; Yang et al., 2021), scene reconstruction (Li et al., 2023; Wang et al., 2023; Guédon & Lepetit, 2024; Yariv et al., 2021), and beyond. Physics-Informed Neural Networks (PINNs) are neural fields that model solutions to differential equations, solved as optimisation problems supervised by physical laws (Raissi et al., 2019; Hao et al., 2022). A wide range of problems, such as solving fluid dynamics (Cai et al., 2021), Eikonal equations (Smith et al., 2021), simulation (Bastek & Kochmann, 2023), and many more, have been achieved with neural fields. Early works that used coordinate-based MLPs struggled to model high-frequency details in signals and derivatives. Siren (Sitzmann et al., 2020) introduces periodic activation functions for MLPs, facilitating gradient supervision with higher-order derivatives, enabling more complex applications (Chen et al., 2023a; Novello et al., 2023; Kairanda et al., 2024). However, Siren and similar global MLP-based methods have shortcomings: they ignore the spatial structure of physical fields, require back-propagating through the full MLP for every coordinate sample, thus leading to slower training times compared to our proposed grid-based representation.

**Feature Grids.** In contrast to coordinate-based MLPs, discrete or hybrid representations learn compact localised features, reducing computation and training time while improving quality. Various discrete representations, such as grids (Sara Fridovich-Keil and Alex Yu et al., 2022; Martel et al., 2021), octrees (Takikawa et al., 2021) and feature planes (Fridovich-Keil et al., 2023; Chan et al., 2022; Chen et al., 2022; Cao & Johnson, 2023), in combination with MLPs, have been introduced for faster learning. Müller et al. (2022) employ a multi-resolution structure with a hash encoding to provide a faster lookup, significantly speeding up computation. Unfortunately, the linear components (linear interpolation, ReLU) of these feature learning methods do not support higher-order derivative supervision required for solving DEs. While some methods (Li et al., 2023; Huang & Alkhalifah, 2024; Wang et al., 2024) support higher-order derivative supervision using finite difference methods, they are known to be approximations and are susceptible to discretisation errors. Chetan et al. (2025) recently introduced a post-hoc differential operator for pre-trained hybrid fields, such as Instant-NGP. Due to the reliance on a pre-trained representation, they fail to solve differential equations. On the other hand, our proposed method combines the advantages of localised grid feature learning with high-order differentiability to enable faster training and convergence.

**Radial Basis Functions.** (Mai-Duy & Tran-Cong, 2003) were among the first to employ RBFs as activation functions, with later works such as (Ramasinghe & Lucey, 2022) and GARF (Chng et al., 2022) further demonstrating the utility of Gaussian activations. Recently, methods inspired by 3D Gaussian Splatting (Kerbl et al., 2023), including Lagrangian Hashing (Govindarajan et al., 2025) and SC-GS (Huang et al., 2024), leverage Gaussian interpolation for feature aggregation, though their focus is on radiance fields rather than solving DEs. The closest to ours is NeuRBF (Chen et al., 2023b), which introduced feature grids with RBF interpolation. However, its objectives and therefore the design differ fundamentally: NeuRBF proposes neural field representations that adapt to known target signals (images, SDFs, or Instant-NGP-distilled NeRFs), whereas $\partial^\infty$-*Grid* solves PDEs from derivative-only supervision without observing the target field; see App. B for details. Their Gaussian interpolation remains discontinuous across grid cells due to the single-ring neighbourhood, and the method requires signal-dependent RBF initialisation with ground-truth signals (or proxy hybrids) in the loss formulation, leading to failure of NeuRBF for our problem setting.

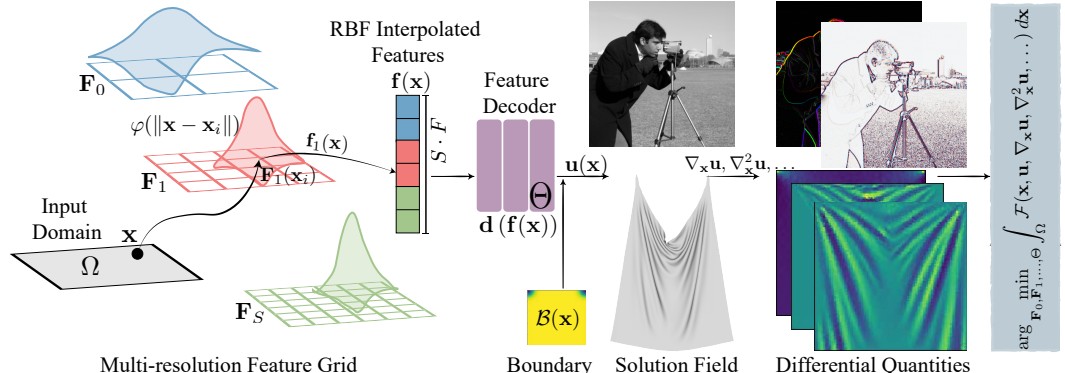

**Figure 2: Illustration of $\partial^\infty$-Grid.** Our grid-based representation accurately represents fields $\mathbf{u} : \Omega \to \mathbb{R}^m$ and efficiently solves differential equations $\mathcal{F}$. We first sample query coordinates $\mathbf{x}$ from the input domain $\Omega$ (here, 2D). We apply infinitely differentiable RBF interpolation $\varphi(\mathbf{x})$ to extract smooth, query-dependent features from the multi-scale learnable feature grids $\{\mathbf{F}_s\}_{s \in [0,\ldots,S]}$. These interpolated features $\{\mathbf{f}_s(\mathbf{x})\}_s$ at different scales are concatenated as $\mathbf{f}(\mathbf{x})$ and then passed through a decoder $\mathbf{d}(\mathbf{f}(\mathbf{x}); \Theta)$ to produce the final signal $\mathbf{u}(\mathbf{x})$ (*e.g.*, image or cloth deformation). The converged neural field (solution) is obtained by optimising the governing equation $\mathcal{F}$ as the loss.

## 3 METHOD

Our goal is to represent and solve for the field $\mathbf{u}$ subject to physical constraints $\mathcal{F}$, expressed as differentiable equations, as shown in Eq. (1). The solution must be accurate, and the optimisation needs to be fast. This section presents our feature grid-based representation, which is computationally efficient (Sec. 3.1). While grid-based representations have been explored in prior work (Müller et al., 2022; Fridovich-Keil et al., 2023), these approaches often lack differentiability with respect to inputs $\mathbf{x}$, making them unsuitable for solving differential equations accurately. In contrast, our method supports higher-order derivatives by combining feature grids with the smoothness and differentiability of RBF interpolation (Sec. 3.2). Additionally, we introduce a multi-resolution technique to effectively capture high-frequency signals which ensures stable optimisation with fast global gradient propagation (Sec. 3.3). An overview of our approach is provided in Fig. 2.

### 3.1 FEATURE GRID FORMULATION

We aim to parameterise the field $\mathbf{u}(\mathbf{x}) : \Omega \to \mathbb{R}^m$, where $\Omega \subset \mathbb{R}^d$ is a low-dimensional ($d = 2, 3$) spatio-temporal input domain. Directly processing the query coordinates $\mathbf{x}$ with a coordinate-based MLP, as in neural fields (Xie et al., 2022; Hao et al., 2022), requires updating all weights during backpropagation, which is computationally expensive and bypasses the opportunity to exploit the local structure of the spatial domain. To address these limitations, we propose a novel encoding-decoding framework. The signal $\mathbf{u}$ is computed as a composition of two functions:

$$\mathbf{u}(\mathbf{x}) = \mathbf{d}(\mathbf{f}(\mathbf{x}; \mathbf{F}); \Theta), \qquad (2)$$

where $\mathbf{f}$ is a feature encoder parametrised by the feature grid $\mathbf{F}$, and $\mathbf{d}$ is a small decoder. This formulation leverages the spatial structure in the feature encoder, requiring only a small set of trainable parameters of grid $\mathbf{F}$—adjacent to the query coordinate $\mathbf{x}$—to be updated during optimisation.

**Feature Encoder.** We adopt a grid-based representation, where the domain of interest is discretised into a regular grid. Each grid node stores a learnable feature vector, which is optimised during training. Unlike traditional coordinate-based MLPs, this representation exploits spatial locality, enabling efficient computation. Since feature grids are discrete, interpolation is required to query continuous points. Formally, let $\mathbf{F} \in \mathbb{R}^{(N+1)^d \times F}$ denote the feature grid, where $N$ is the resolution (assumed identical along all spatial dimensions $d$ for simplicity), and $F$ is the feature dimension at each grid node. Given a query coordinate $\mathbf{x} \in \mathbb{R}^d$, the corresponding feature vector $\mathbf{f}(\mathbf{x}) \in \mathbb{R}^F$ is obtained via interpolation over the grid. The interpolated feature is computed as:

$$\mathbf{f}(\mathbf{x}) = \sum_{i \in \mathcal{N}(\mathbf{x})} w(\mathbf{x}, \mathbf{x}_i) \mathbf{F}(\mathbf{x}_i), \qquad (3)$$

where $\mathcal{N}(\mathbf{x})$ is the set of neighbouring grid nodes enclosing $\mathbf{x}$; $\mathbf{F}(\mathbf{x}_i)$ and $w(\mathbf{x}, \mathbf{x}_i)$ are the learnable feature vector and the scalar interpolation weight corresponding to node $\mathbf{x}_i$, respectively.

**Feature Decoder.** To map the interpolated feature $\mathbf{f}(\mathbf{x})$ to the target signal $\mathbf{u}(\mathbf{x})$, we employ a feature decoder $\mathbf{d}(\cdot; \Theta) : \mathbb{R}^F \rightarrow \mathbb{R}^m$ with weights $\Theta$. The decoder can be implemented as either a simple linear layer or a small MLP (with smooth activation *e.g.*, $\tanh$), depending on the complexity of the task. This additional decoding step allows the model to map the features to the target flexibly.

**Differential Equation Solver.** We propose to compute gradients, Jacobians, and Laplacians of the decoded interpolated features with respect to query samples $\mathbf{x}$ using automatic differentiation (autograd). These derivatives are essential for solving differential equations. Recalling our goal of optimising for $\mathbf{u}$ in problems of the form Eq. (1), we define the loss function as:

$$\mathcal{L}(\mathbf{F}, \Theta) = \int_\Omega \mathcal{F}(.) \, d\mathbf{x} = \sum_{i \in \mathcal{D}} \mathcal{F}\left(\mathbf{x}_i, \mathbf{u}(\mathbf{x}_i), \nabla_{\mathbf{x}_i} \mathbf{u}(\mathbf{x}_i), \nabla^2_{\mathbf{x}_i} \mathbf{u}(\mathbf{x}_i), \dots; g(\mathbf{x}_i)\right), \tag{4}$$

where $\mathbf{u}(\mathbf{F}, \Theta)$ is parameterised as in Eq. (2). The loss function represents the PDE residual over the spatio-temporal domain, optionally including boundary constraints or data terms. Our framework accommodates various formulations: strong form (*e.g.*, Poisson), variational form with boundary constraints (*e.g.*, cloth simulation), and data-driven terms (*e.g.*, SDF). To guarantee uniqueness and stability, we typically impose Dirichlet boundary or initial conditions as hard constraints, which are known to improve convergence (Kairanda et al., 2024; Lu et al., 2021). This is achieved by modulating the predicted field in Eq. (2) with a boundary-aware function:

$$\mathbf{u}(\mathbf{x}) = \mathbf{d}(\mathbf{f}(\mathbf{x}; \mathbf{F}); \Theta) \, \mathcal{B}(\mathbf{x}) + \mathbf{h}(\mathbf{x})\left(1 - \mathcal{B}(\mathbf{x})\right), \tag{5}$$

where $\mathbf{h}$ denotes the known boundary function and $\mathcal{B}(\mathbf{x}) = 1 - e^{-\|\mathbf{x} - \mathbf{x}_{\partial\Omega}\|^2 / \sigma}$ is a distance-based weighting that enforces $\mathbf{u}(\mathbf{x}) = \mathbf{h}(\mathbf{x})$ for all $\mathbf{x} \in \partial\Omega$. This formulation is equally applicable to initial conditions, where the boundary is defined by $t = 0$.

During training, we optimise both the feature grid $\mathbf{F}$ and the decoder $\Theta$ using gradient-based optimisation. In practice, the integral in Eq. (4) is estimated by sampling points from the domain $\Omega$. A dataset $\mathcal{D} = \{(\mathbf{x}_i, g(\mathbf{x}_i))\}_i$ is constructed by stratified random sampling of coordinates $\mathbf{x}_i \in \Omega$ for PDE evaluation, optionally including data coordinates for loss terms and their associated known quantities $g(\mathbf{x}_i)$. The dataset $\mathcal{D}$ is sampled dynamically during training, progressively refining the approximation of $\mathcal{L}$ as the number of samples increases.

## 3.2 RADIAL BASIS FUNCTION INTERPOLATION

In Eq. (3), the interpolation weights $w(\mathbf{x}, \mathbf{x}_i)$ play a crucial role in determining the contribution of neighbouring grid points to the interpolated features. These weights can be computed using various methods. Prior works on scene representations often rely on $d$-linear interpolation (Müller et al., 2022; Fridovich-Keil et al., 2023), where the weights are determined based on the relative distances between the query point $\mathbf{x}$ and the nearest grid points along each dimension. Formally, let $\mathbf{x} = (x_1, x_2, \dots, x_d)$ represent the query point, and $\mathbf{x}_i = (x_{i1}, x_{i2}, \dots, x_{id})$ denote a grid node. The weight for $d$-linear interpolation is given by:

$$w(\mathbf{x}, \mathbf{x}_i) = \prod_{k=1}^d \left(1 - \frac{|x_k - x_{ik}|}{\sigma}\right), \tag{6}$$

where $\sigma$ is the grid cell size. While linear interpolation is computationally efficient, it is only $C^0(\mathbb{R})$ (continuous but not differentiable) at grid nodes, and $C^1(\mathbb{R})$ in the grid interior, when differentiating with respect to query coordinates. The lack of support for higher-order derivatives prohibits the computation of differential quantities, such as Jacobians and Laplacians, which are essential for solving DEs as described in Eq. (4). An alternative to linear interpolation is to employ higher-order basis functions, such as quadratic, cubic, or RBFs. We employ RBF interpolation (Wright, 2003), which is smooth across neighbourhoods and infinitely differentiable ($C^\infty(\mathbb{R})$). The interpolation weights are defined as

$$w(\mathbf{x}, \mathbf{x}_i) = \frac{\varphi(\|\mathbf{x} - \mathbf{x}_i\|)}{\sum_{j \in \mathcal{N}(\mathbf{x})} \varphi(\|\mathbf{x} - \mathbf{x}_j\|)}, \tag{7}$$

where $\mathcal{N}(\mathbf{x})$ denotes the neighbourhood of $\mathbf{x}$, and the denominator normalises the weights. We use Gaussian RBF kernels, $\varphi(r) = e^{-(\varepsilon r)^2}$, where $\varepsilon$ controls the kernel width. RBFs provide a general-purpose, differentiable representation that supports higher-order derivatives when required. For instance, fourth-order PDEs such as the Euler–Bernoulli beam or Kirchhoff–Love plate equations (Guo et al., 2021) demand differentiability beyond what cubic bases can provide, particularly in strong-form settings. Moreover, RBFs offer tunable support via the shape parameter $\varepsilon$ (which we set as a hyperparameter), enabling a flexible trade-off between locality and accuracy that fixed-degree polynomials cannot achieve.

**Efficient RBF Interpolation.** In RBF interpolation, neighbouring grid points $\mathcal{N}(\mathbf{x})$ are weighted according to their distance from the query point $\mathbf{x}$. As shown in Eqs. (3) and (7), Gaussian RBFs $(\varphi(r) = e^{-(\varepsilon r)^2})$ are globally supported, requiring the neighbourhood to be the full grid—leading to high memory and computational costs. Consequently, standard RBF interpolation becomes impractical for high-resolution grids or large numbers of samples.

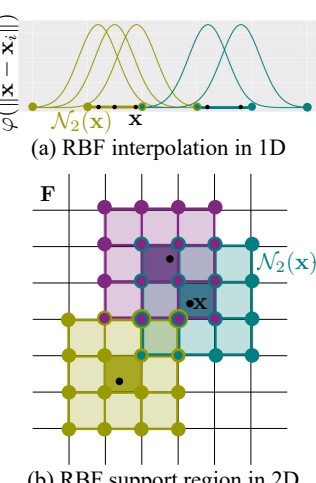

(a) RBF interpolation in 1D

(b) RBF support region in 2D

To match the efficiency of linear interpolation with the smoothness of RBFs, we sample the domain in a structured way and precompute neighbourhood indices. At each training step, we stratify samples $\mathbf{x}$ over a domain $\Omega$, ensuring they are both uniformly distributed and random for a more accurate approximation of Eq. (4).

In conjunction with stratified sampling, we propose precomputation of the neighbourhood $\mathcal{N}(\mathbf{x})$, which can be reused across training iterations. For a given query point $\mathbf{x}$, the effective neighbourhood $\mathcal{N}_\rho(\mathbf{x})$, where the RBF weights are non-zero, is determined by the shape parameter $\varepsilon$ of the RBF kernel. For example, if $\varepsilon = 2$, only the 1-ring neighbourhood $\mathcal{N}_1(\mathbf{x})$ needs to be considered, whereas for $\varepsilon = 1$, the 2-ring neighbourhood $\mathcal{N}_2(\mathbf{x})$ is required. Since the sample point may lie anywhere within a grid

**Figure 3:** Illustration of effective neighbourhoods $\mathcal{N}_2(\mathbf{x})$ in $d = 1, 2$, determined by the shape parameter $\varepsilon$. An additional ring accounts for stratified sampling.

cell, including near boundaries, we include an additional ring in practice. For instance, $\mathcal{N}_2(\mathbf{x})$ is used for $\varepsilon = 2$; see Fig. 3 for an illustration in $d = 1, 2$. This approach ensures smooth interpolation with feature overlap across query samples, ensuring accurate learning of the feature grid.

## 3.3 MULTI-RESOLUTION FEATURE GRID

We observe that our feature encoder with a single-resolution grid captures local gradients effectively and requires many time steps to propagate gradient information across the entire domain. This limitation stems from the small interpolation neighbourhoods used in relatively large feature grids. Drawing inspiration from recent advances in scene representations (Müller et al., 2022; Fridovich-Keil et al., 2023), we propose a multi-resolution approach to overcome this challenge. We define a set of multi-scale feature grids, each representing a different resolution. Formally, for a given the base grid resolution $N_{\max}$ and the number of scales $S$, the feature grids $\{\mathbf{F}_s\}_{s \in [0,\ldots,S-1]}$ are learnable parameters: $\mathbf{F}_s \in \mathbb{R}^{(N+1)^d \times F}$ where $N = N_{\max}/2^s$ is the resolution at scale $s$. With this formulation, the interpolated features from all scales are concatenated to form a multi-scale feature vector $\mathbf{f}(\mathbf{x}) = (\mathbf{f}_0(\mathbf{x}), \mathbf{f}_1(\mathbf{x}), \ldots, \mathbf{f}_{S-1}(\mathbf{x})) \in \mathbb{R}^{S \cdot F}$, where $\mathbf{f}_s(\mathbf{x})$ is computed using the RBF interpolation described in Eqs. (3) and (7).

The multi-resolution feature grid offers several advantages. First, it is computationally efficient, as the hierarchical structure reduces the cost of global gradient propagation. Second, it is highly expressive, as the combination of coarse and fine features, along with their gradients, enables the stable and accurate recovery of complex signals without requiring adaptive RBF shapes.

## 4 EXPERIMENTS

We experimentally demonstrate the versatility of our representation across diverse signal types, including the Poisson equation (Sec. 4.1), the Helmholtz equation (Sec. 4.2), and neural cloth simulation (Sec. 4.3). Unlike prior grid-based methods such as Instant-NGP (Müller et al., 2022),

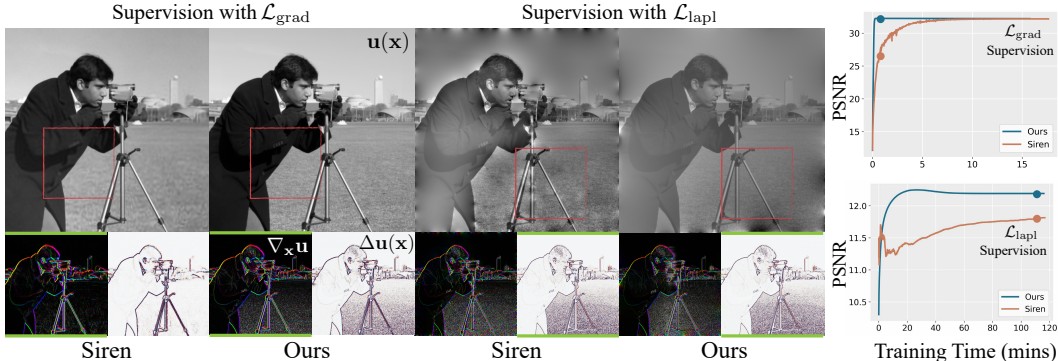

**Figure 4: Image Reconstruction.** Comparison of $\partial^\infty$-*Grid* with Siren (Sitzmann et al., 2020) for the reconstruction from gradient and Laplacian fields (Poisson equation). Results are visualised at training time marked by "•" and "•", and the supervision signal is highlighted with a green border. Our method achieves higher PSNR with faster convergence for high-resolution images, while Siren struggles, especially for Laplacian.

**Table 1:** Comparison against Siren (Sitzmann et al., 2020) and K-planes (Fridovich-Keil et al., 2023) for image reconstruction (Fig. 4). $\partial^\infty$-*Grid* achieves higher PSNR with substantially reduced training time than Siren. K-planes struggle (for gradient supervision) or fail (for Laplacian) due to non-differentiable interpolation.

| Model | K-Planes | | Siren | | Ours | |
|---|---|---|---|---|---|---|
| Supervision | Grad. | Lapl. | Grad. | Lapl. | Grad. | Lapl. |
| PSNR | 17.96 | - | 32.12 | 11.82 | 32.24 | 12.19 |
| Training time | 9.5mins | - | 10mins | 1hr56mins | 25s | 15mins |
| Parameters | 8.4m | - | 329.99k | 1.32m | 702.94k | 700.81k |

K-Planes (Fridovich-Keil et al., 2023), and NeuRBF (Chen et al., 2023b), our feature grid representation accurately solves for fields. We show that $\partial^\infty$-*Grid* achieves significant speed-ups over sinusoidal neural networks (Sitzmann et al., 2020) while maintaining comparable accuracy, and that it can recover high-frequency solution fields where PINNs struggle (see Sec. B). Apart from solving DEs, we demonstrate that our representation supports direct signal fitting (Sec. 4.4). We also include ablations on the RBF kernel size and the multiscale design of the feature grid (Sec. 4.5). Additional experiments are provided in the appendices, *i.e.*, SDF solutions to the Eikonal equation, the spatio-temporal heat equation, and 1D/2D advection with Gaussian waves (with comparison to (Chen et al., 2023a)). We further demonstrate highly challenging cases, such as Zalesak's disk and Helmholtz problems with higher wavenumbers.

## 4.1 IMAGE RECONSTRUCTION AS POISSON EQUATION SOLVE

To demonstrate that our feature grid representation can accurately reconstruct a field and its derivatives, we first consider a simple differentiable equation—the Poisson equation. This equation represents a class of problems where the objective is to estimate an unknown signal $g$ from discrete samples of its Laplacian $\Delta g = \nabla \cdot \nabla g$. As a simpler variant, we additionally estimate the signal from its gradients $\nabla g$, which can be formulated as minimising one of the following losses (as an instance of Eq. (4)):

$$\mathcal{L}_{\text{grad}} = \int_\Omega \|\nabla_{\mathbf{x}} \mathbf{u}(\mathbf{x}) - \nabla_{\mathbf{x}} g(\mathbf{x})\| \, d\mathbf{x}, \quad \mathcal{L}_{\text{lapl}} = \int_\Omega \|\Delta \mathbf{u}(\mathbf{x}) - \Delta g(\mathbf{x})\| \, d\mathbf{x}. \tag{8}$$

We solve the Poisson equation for images by supervising the model solely with higher-order derivatives—either ground-truth gradients, as in $\mathcal{L}_{\text{grad}}$, or Laplacians $\mathcal{L}_{\text{lapl}}$. Our model successfully reconstructs the target image, as shown in Figs. I, 4, IV and 7. For colour images we supervise gradients channel-wise and visualise the red-channel field for clarity. Global intensity variations remain due to the ill-posedness of the inverse problem, as boundary conditions are not explicitly specified.

To assess computational efficiency against coordinate-based MLPs, we evaluate high-resolution image reconstruction with our method and compare it to Siren (Sitzmann et al., 2020). While the

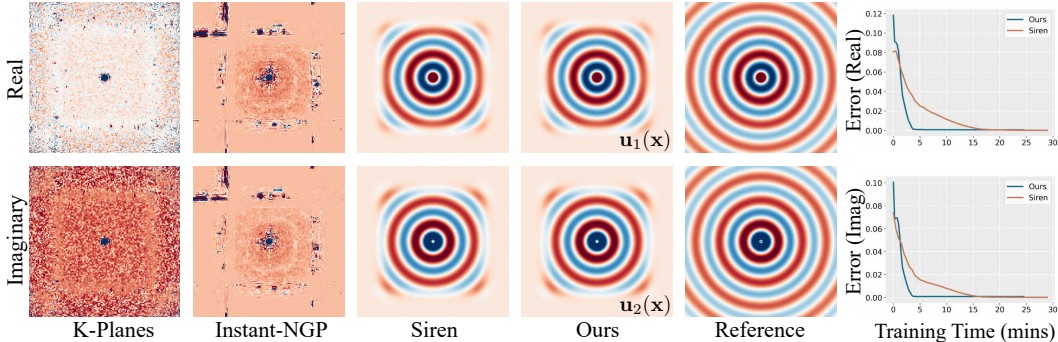

**Figure 5: Helmholtz Equation.** Comparison for solving the Helmholtz equation (second-order PDE) for a single point source at the centre. Prior hybrid architectures, such as K-Planes (Fridovich-Keil et al., 2023) and Instant-NGP (Müller et al., 2022), break down as their linear components equate to zero for second-order derivatives. Our method matches reference with significantly faster training time compared to Siren, as shown in the $\ell_1$ error plot on the right and $\ell_2$-error visualisations in Fig. VI.

original Siren paper reports results on $256 \times 256$ images, we use the full $512 \times 512$ resolution for a fairer comparison and clearer demonstration of training speed. We adopt the smallest Siren architecture that converges reliably, as reducing capacity can speed up training but could also lead to underfitting. As shown in Fig. 4-(right), our method captures both global structure and fine local detail more accurately—and converges significantly faster ($20\times$). Quantitative comparisons for the Poisson equation are reported in Tab. 1 for our method, Siren, and K-Planes (Fridovich-Keil et al., 2023). Notably, existing grid-based approaches such as K-Planes cannot handle higher-order supervision: with linear interpolation, first derivatives are discontinuous at grid boundaries and second derivatives vanish everywhere, preventing recovery from Laplacian fields.

## 4.2 HELMHOLTZ EQUATION

We further demonstrate the applicability of our method to wave propagation problems governed by complex boundary conditions. Specifically, we consider the Helmholtz equation, a second-order partial differential equation commonly used to model steady-state wave behaviour and diffusion phenomena. The Helmholtz equation is given by:

$$H(m)\,\mathbf{u}(\mathbf{x}) = -g(\mathbf{x}), \quad \text{with} \quad H(m) = \left(\Delta + m(\mathbf{x})\,\omega^2\right), \tag{9}$$

where $g(\mathbf{x})$ denotes a known source function, $\mathbf{u}(\mathbf{x})$ is the unknown complex-valued wavefield, $\omega$ is the wave number, and $m(\mathbf{x}) = \frac{1}{c(\mathbf{x})^2}$ represents the squared slowness as a function of wave velocity $c(\mathbf{x})$. To solve for the wavefield, we parameterise $\mathbf{u}(\mathbf{x})$ using our proposed $\partial^\infty$-*Grid* representation. The decoder outputs two channels corresponding to the real and imaginary components of the wavefield. We set up the input domain as $\Omega = \left\{\mathbf{x} \in \mathbb{R}^2 \,\middle|\, \|\mathbf{x}\|_\infty < 1\right\}$, and consider a spatially uniform wave velocity with a single point source, modelled as a Gaussian with variance $\sigma^2 = 10^{-4}$. The model is trained using a loss function derived directly from the Helmholtz formulation:

$$\mathcal{L}_{\text{Helmholtz}} = \int_\Omega \lambda(\mathbf{x})\, \|H(m)\,\mathbf{u}(\mathbf{x}) + g(\mathbf{x})\|_1 \, d\mathbf{x}, \tag{10}$$

where $\lambda(\mathbf{x}) = k$ (a tunable hyperparameter) for locations where $g(\mathbf{x}) \neq 0$, corresponding to the inhomogeneous part of the domain, and $\lambda(\mathbf{x}) = 1$ elsewhere. Unlike the image reconstruction task from derivative fields—where data sampling is required—we employ stratified sampling here, since $\mathcal{L}_{\text{Helmholtz}}$ can be evaluated at arbitrary points within the domain. We incorporate perfectly matched layers (PML), which modify the Helmholtz PDE to attenuate outgoing waves near the boundary, ensuring uniqueness of the solution (see Sec. E for implementation details). All methods, including ours, predict only the interior of the domain, as the PML-modified Helmholtz formulation absorbs steady-state wavefields generated by the point source. The reference solution is computed in closed form using Hankel functions and therefore extends across the entire domain.

Our representation achieves high-fidelity reconstructions of the wavefield, as shown in Figs. 1 and 5 for $\omega = 20$, where we compare against the closed-form solution. While both methods match the

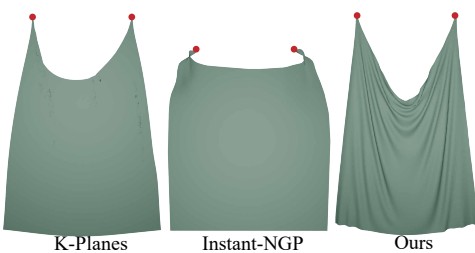

**Figure 6:** Kirchhoff-Love Simulation (Kairanda et al., 2024). $\partial^\infty$-*Grid* captures fine-grained cloth wrinkles by solving a highly nonlinear problem.

**Table 2:** Ablation study for varying RBF shapes $\varepsilon$ and neighbourhood sizes $\rho$ for the high-resolution gradient-based image reconstruction in Fig. I.

| Shape $\varepsilon$ | Ring $\rho$ | PSNR ↑ | SSIM ↑ | Training ↓ |
|---|---|---|---|---|
| 0.6 | 4 | 28.74 | 0.87 | 15 min |
| 1 | 4 | 28.74 | 0.87 | 17 min |
| 1 | 3 | 28.73 | 0.87 | 12 min |
| 1 | 2 | 28.82 | 0.86 | 12 min |
| 1* | 1 | 27.50 | 0.82 | 15 min |
| 2* | 1 | 17.72 | 0.74 | 6 min |

reference closely, our approach converges faster, yielding a $4\times$ speed-up over Siren; see Fig. 5-(right). Additional results for more challenging cases with $\omega = 30, 40$ are presented in Sec. E and Tab. I. We also compare with prior feature grid methods: (Fig. 5-(left)): They fail as effective solvers since they cannot capture the higher-order derivatives required by the Helmholtz equation.

### 4.3 Cloth Simulation

NeuralClothSim (Kairanda et al., 2024) introduces a quasistatic cloth simulation framework that models surface deformations as a coordinate-based implicit neural deformation field (NDF). Given a simulation setup—defined by the cloth's initial configuration, material parameters, boundary constraints and applied external forces—the method learns the equilibrium deformation field governed by the Kirchhoff-Love thin shell theory. The deformation is represented as an NDF $\mathbf{u}(\mathbf{x}) : \Omega \to \mathbb{R}^3$, where $\Omega \subset \mathbb{R}^2$ denotes the cloth's parametric domain. In the context of volumetric thin shells such as cloth, the Kirchhoff-Love model (Wempner & Talaslidis, 2003) provides a reduced kinematic formulation, where the shell midsurface fully characterises the strain distribution across the thickness. Following standard thin shell assumptions, the deformation field $\mathbf{u}$ is used to compute the Green strain, which is decomposed into membrane strain $\varepsilon$ and bending strain $\kappa$, capturing in-plane stretching and out-of-plane curvature changes, respectively. The internal hyperelastic energy $\Psi[\varepsilon, \kappa; \mathbf{\Phi}]$ is then formulated as a functional of these geometric strains and the material properties $\mathbf{\Phi}$. Under the action of external forces $\mathbf{g}$ and boundary conditions $\mathbf{b}$, the simulation seeks an equilibrium configuration by minimising the total potential energy, combining internal energy and external work. This principle is implemented by defining the loss as (see Sec. F for further details):

$$\mathcal{L}_{\text{cloth}} = \int_\Omega \Psi[\varepsilon(\mathbf{u}(\mathbf{x})), \kappa(\mathbf{u}(\mathbf{x})); \mathbf{\Phi}] - \mathbf{g}(\mathbf{x}) \cdot \mathbf{u}(\mathbf{x}) \, d\mathbf{x}, \quad \text{subject to } \mathbf{u}(\mathbf{x}) = \mathbf{b}(\mathbf{x}) \text{ on } \partial\Omega. \quad (11)$$

Note that Kairanda et al. (2024) employ Siren as the representation for NDF $\mathbf{u}$, while we use our $\partial^\infty$-*Grid*. As a specific example, we consider an analytically defined square piece of cloth of mass density $\rho$ held with two handles and subject to a gravity force $\mathbf{g} = [0, 0, -9.8\rho]$. We incorporate Dirichlet constraints directly into the loss following Eq. (5), and assume a linear elastic material model for $\mathbf{\Phi}$. See Fig. 6 for results, where we recover realistic simulations, including folds and wrinkles, by solving a highly non-linear PDE where stretching and bending strains depend non-linearly on the deformation. Additional comparisons with K-Planes and Instant-NGP show their failure, as they are unable to model the membrane and bending strains, which require higher-order derivatives of the deformation field. As multiple equilibria are possible for the cloth deformation (thus, no single reference solution), we exclude cloth simulation from the numerical accuracy evaluation.

### 4.4 Signal Representation

The following additional experiment underscores the versatility of our representation (although not being the central focus of our work). Beyond solving DEs, our method can also fit signals directly by instantiating Eq. (4) with a supervision signal, *i.e.*,

$$\mathcal{L}_{\text{signal}} = \int_\Omega \|\mathbf{u}(\mathbf{x}) - \mathbf{g}(\mathbf{x})\| \, d\mathbf{x}, \quad (12)$$

where $\mathbf{g}(\mathbf{x})$ denotes the known target signal such as a 2D image or an SDF. We showcase both cases in Fig. 7-(a,c), where our method preserves high-frequency details on par with previous competitive

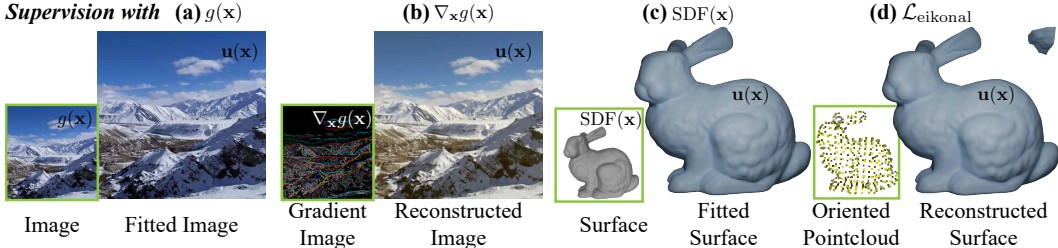

**Figure 7: Results of signal representation.** Our $\partial^\infty$-*Grid* matches high-frequency details when fitting images and SDFs directly (a,c) and, unlike prior grid methods (Müller et al., 2022; Chen et al., 2023b) (Figs. I and V and Tab. 1), also succeeds with gradient and Eikonal supervision (b,d). Inputs are outlined in green.

grid-based baselines (Müller et al., 2022; Chen et al., 2023b). Our representation further supports indirect supervision: We recover images from gradient fields and SDFs from oriented point clouds by solving the Eikonal equation (Sec. C.0), as illustrated in Fig. 7-(b,d). In these settings, the competing grid-based methods fail (Figs. I and V and Tab. 1 in the Appendix). Together, these results show that a single differentiable grid accommodates both direct signal fitting and PDE-constrained reconstruction without architectural changes.

## 4.5 ABLATIONS AND FURTHER COMPARISONS

**Neighbourhood Ring Selection.** The choice of RBF shape parameter $\varepsilon$, and consequently the effective neighbourhood $\mathcal{N}_\rho$, balances computational efficiency and accuracy. To study the trade-off between truncation, accuracy, and speed—as well as potential instabilities—we conducted ablations on image recovery from the higher-resolution $768 \times 768$ gradient image in Fig. I. First, we vary $\varepsilon$ and adjust the neighbourhood size adaptively, so that the RBF weights are non-zero (*e.g.*, $\rho = 4$ for $\varepsilon = 0.6$). Smaller $\varepsilon$ values with larger rings produce smoother, more accurate reconstructions but at higher computational cost, while overly narrow kernels (*e.g.*, $\varepsilon = 2$, $\rho = 1$) introduce discontinuities resembling linear interpolation. Second, we fixed $\varepsilon = 1$ and varied the ring size $\rho$. The 1-ring setting showed visible artefacts, whereas $\varepsilon = 1$ with a 2- or 3-ring neighbourhood offers stable training and smooth interpolation without artefacts, matching our intuition from Fig. 3.

**Multi-resolution Grid.** We demonstrate the effectiveness of multi-resolution feature grids in Fig. II. In the SDF example, the single-resolution grid restricts the Eikonal loss to local regions, leading to noticeable artefacts. In contrast, the multi-resolution grids enable gradient propagation across the entire domain while incurring minimal additional parameter overhead. Next, as illustrated by the image recovery example (Fig. II), this design leads to significantly faster convergence for PSNR.

**Comparisons to NeuRBF (Chen et al., 2023b).** We modified NeuRBF's image-fitting objective to use our gradient-based Poisson loss and retained their gradient-weighted RBF initialisation; the reconstruction failed to converge at PSNR/SSIM $10.85/0.17$ with visible artefacts (cf. Fig. I). Applying the same pipeline to Eikonal SDF fitting from oriented point clouds also failed to recover a smooth surface (cf. Fig. V), highlighting the difficulty of training NeuRBF under purely differential supervision. Please refer to App. B for the detailed analysis.

## 5 CONCLUSION

The experiments demonstrate that $\partial^\infty$-*Grid* enables computation of higher-order derivatives directly from the interpolated features, thanks to smooth and infinitely differentiable RBF interpolation weights. We can reliably compute gradients, Jacobians, and Laplacians for solving PDEs. This leads to accurate results for Poisson and Helmholtz equations, allows recovery of high-frequency fields for challenging and non-linear PDEs, such as the Kirchhoff-Love boundary value problem for cloth simulation and the Eikonal equation. Moreover, we experimentally show faster convergence compared to coordinate-based MLPs. One current limitation is that the interpolation becomes expensive for higher-dimensional inputs, similar to numerical solvers, and a potential solution for that in the future could be projection to lower-dimensional spaces. Moreover, while we observed convergence across all experiments with the proposed RBF neighborhood truncation, further investigation is needed to establish theoretical guarantees for $C^\infty$ under truncation.

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

APPENDICES

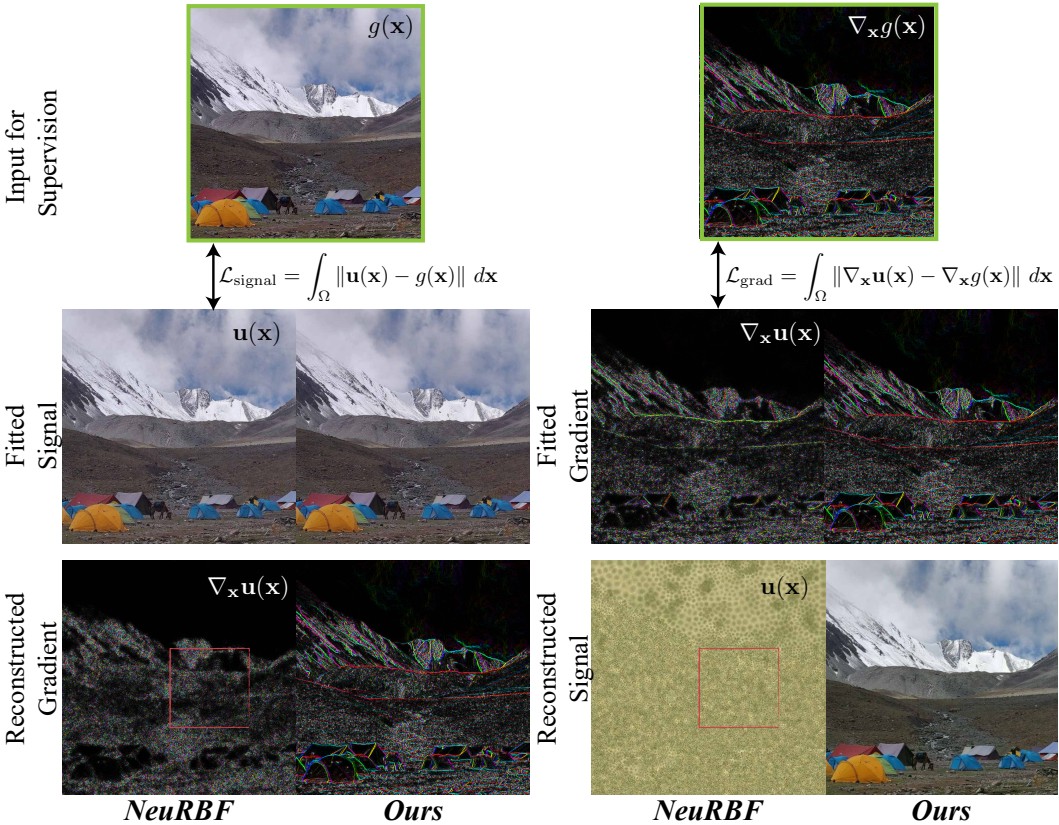

**Figure I: Comparisons to NeuRBF (Chen et al., 2023b)** for fitting 2D images with direct image supervision (left) and gradient image supervision (right). Note that while NeuRBF can fit signals well, it fails to compute accurate gradients (bottom left) and, consequently, cannot recover images with gradient supervision.

## A    REPRODUCIBILITY DETAILS

$\partial^\infty$-**Grid Implementation.** Our method is implemented in PyTorch, and all our experiments and the speed comparison for Siren are conducted on a single H100 GPU. Following (Müller et al., 2022), each feature grid is initialised with small random values drawn from a uniform distribution, *i.e.* $\mathbf{F}_s \sim \mathcal{U}(-10^{-4}, 10^{-4})$, whereas the decoder (whether linear or MLP) weights are initialised with Xavier (Glorot & Bengio, 2010) and biases are set to zero. In the case of MLP decoder, we employ $\tanh$ as the activation. A learning rate of $5 \times 10^{-3}$ is used with the Adam optimiser. We set $\varepsilon = 1$, which selects the neighbourhood $\mathcal{N}_3(\mathbf{x})$ (*i.e.*, $\rho = 3$) around each query point $\mathbf{x}$. Because a ring $\mathcal{N}_\rho(\mathbf{x})$ contains $(2\rho)^d$ neighbours per scale, the Gaussian RBF therefore uses $6^d$ neighbours at every scale/grid. Neighbourhood indices $\mathcal{N}_3(\mathbf{x}_i)$ are precomputed for all training and test samples $\{\mathbf{x}_i\}_i$. Additional hyperparameters, including the scale $S$, grid resolution $N_{\max}$, and feature dimension $F$, are experiment specific and detailed in the respective section for reproducibility.

**K-Planes Comparison.** We provide the settings used to reproduce our K-Planes results with the official implementation (Fridovich-Keil et al., 2023) on neural cloth simulation (Sec. 4.3) and Helmholtz equation (Sec. 4.2). We extend K-Planes using the same losses as in our method. The architectures are summarised as follows:

- NeuralClothSim: single-resolution 2D grid of size $64 \times 64$ with feature dimension 16, followed by an MLP with two 64-wide ReLU layers and a linear head; learning rate $10^{-2}$ with 0.1 decay every 2000 steps for 10000 epochs.
- Helmholtz: single-resolution 2D grid of size $128 \times 128$ with feature dimension 32, followed by an MLP with two 64-wide ReLU layers and a linear head; learning rate $2 \times 10^{-5}$ for 50000 epochs. We employ single resolution as the multi-grid K-Planes struggles to converge otherwise.

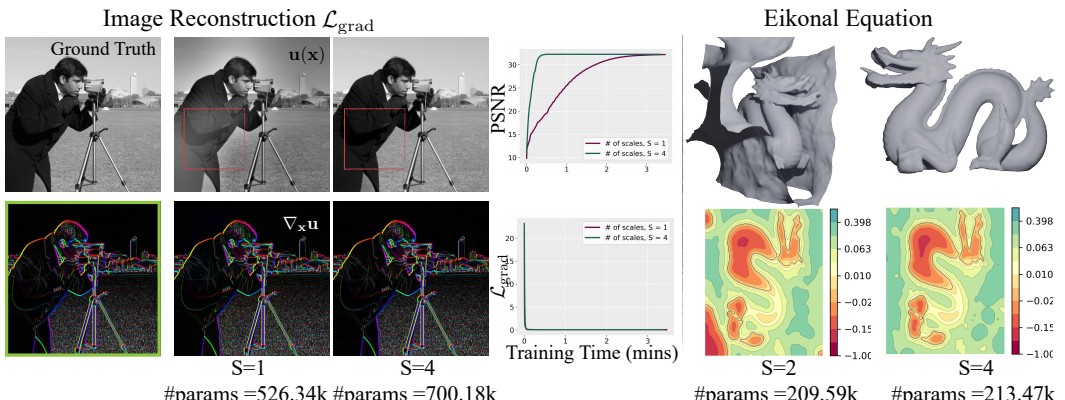

**Figure II: Effectiveness of the multi-resolution grids.** While the single-resolution feature encoder converges at a similar rate to the multi-resolution grid in terms of the differential equation loss (as shown in the gradient loss plot), the $\partial^\infty$-*Grid* multi-resolution architecture enables faster and more accurate field reconstruction (as illustrated in the PSNR plot, achieves higher PSNR in fewer iterations). The image reconstruction results are visualised at $\approx$30 seconds of training.

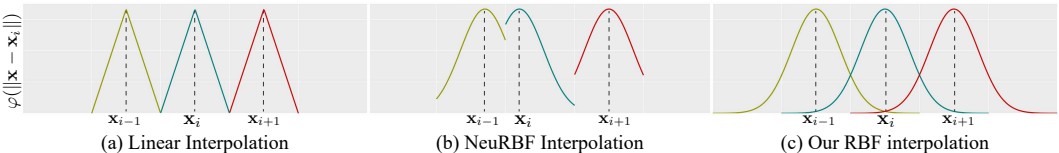

**Figure III: Comparison of interpolation schemes.** Linear interpolation in Instant-NGP/K-Planes (Müller et al., 2022; Fridovich-Keil et al., 2023) is only $C^0$ at grid nodes $\mathbf{x}_i$, whereas our RBF interpolation is $C^\infty$ because centres are fixed on the grid with overlapping neighbourhoods. NeuRBF (Chen et al., 2023b) also uses RBFs but places adaptive centres off-grid and restricts each query to a single neighbourhood, leading to discontinuities.

**Instant-NGP Comparison.** We use the PyTorch implementation (Tang, 2022) of Instant-NGP (that supports autograd) and apply the same losses as in our method. The architecture details are as follows:

- NeuralClothSim: eight-level hash grid (base resolution 4, final resolution 64) with feature dimension 2, followed by a linear head; learning rate $10^{-4}$ with 0.1 decay every 1000 steps for 10000 epochs.
- Helmholtz equation: eight-level hash grid (base resolution 4, final resolution 128) with feature dimension 2, followed by an MLP with two 64-wide ReLU layers and a linear head; learning rate $2 \times 10^{-5}$ for 50000 epochs.
- SDF fitting: sixteen-level hash grid (base resolution 16, final resolution 2048) with feature dimension 2, followed by an MLP with two 64-wide ReLU layers and a linear head; learning rate $10^{-4}$ for 20 epochs.

# B  FURTHER COMPARISONS

## B.1  COMPARISON WITH NEURBF (CHEN ET AL., 2023B)

NeuRBF and $\partial^\infty$-*Grid* both use RBF-interpolated feature grids, yet they target very different problems. NeuRBF focuses on signal fitting: it reconstructs images and SDFs from ground-truth signals and tackles NeRFs via distillation, always coupled with a hybrid backbone such as Instant-NGP, K-Planes, or TensoRF. It does not attempt to solve PDEs directly from differential constraints, which is precisely our setting.

Consequently, NeuRBF assumes access to the full target signal (or a proxy provided by the hybrid model), whereas our losses (Eqs. (8) and (13)) only require derivative supervision. We recover solu-

tions from gradients, Laplacians, or oriented point clouds without seeing the ground-truth field. For images and SDFs, NeuRBF augments its adaptive RBFs with Instant-NGP features for $C^0$ continuity; for NeRF, it first warms up K-Planes or TensoRF for 1–2k steps and then distils their predictions to initialise the RBFs. Such proxy supervision is infeasible in our PDE setting because Instant-NGP/K-Planes themselves fail when trained with purely differential signals.

NeuRBF chooses adaptive RBFs to position and scale kernels based on the known signal, while we adopt fixed-shape Gaussian RBFs to guarantee smooth, differentiable interpolation suitable for PDE losses. Even if one tried to adapt NeuRBF to PDEs, several critical problems remain that we summarise below:

- **Signal-dependent initialisation.** NeuRBF requires hand-crafted initialisation of RBF centres and shapes via weighted $k$-means. This biases the RBF distribution towards data points with higher weights (making their RBFs adaptive). Weights depend on the GT signal: $w_i = \|\nabla I(\mathbf{x}_i)\|$ for images, $w_i = 1/(|\text{SDF}(\mathbf{x}_i)| + 10^{-9})$ for SDFs. For NeRF, NeuRBF first warms up K-Planes or TensoRF for 1–2k steps and distils their predictions to initialise the RBFs as $w_i = (1 - \exp(-\sigma(\mathbf{x}_i)\delta))\|\nabla \mathbf{f}_c(\mathbf{x}_i)\|$. Such proxy supervision is infeasible for PDEs, because Instant-NGP/K-Planes themselves fail in our settings, and no ground-truth field is available. By contrast, we place fixed-shape Gaussian RBFs on grid nodes without bespoke initialisation, which allows us to additionally precompute the effective neighbourhood; see Fig. III for RBF placement of ours vs NeuRBF.
- **Discontinuities in interpolation.** NeuRBF restricts interpolation to a single RBF ring, causing sharp discontinuities whenever the active neighbour set changes; they lean on Instant-NGP to enforce $C^0$ continuity at grid nodes (Fig. III), which is insufficient for PDE supervision. Expanding the neighbourhood is non-trivial because it would require KD-tree lookups at every query point as RBF centres change, and their signal-dependent initialisation scheme is provided only for the single-ring case. PDE losses such as the Eikonal constraint demand smooth gradients across cell boundaries, so we precompute overlapping $\mathcal{N}_3$ neighbourhoods on the grid for arbitrary query points, guaranteeing continuous interpolation and stable higher-order derivatives (Figs. 3 and III).

**Empirical comparison.** To check if NeuRBF can handle PDE supervision, we modify the official NeuRBF implementation to (i) replace its image-fitting objective with our gradient-based Poisson loss (Eq. (8)) and (ii) optimise the Eikonal loss (Eq. (13)) for SDF reconstruction from oriented point clouds. We keep their initialisation scheme—weighting $k$-means by gradient magnitudes for images and by inverse SDF magnitudes for point clouds—yet note that no analogous scheme exists for higher-order PDE supervision (*e.g.*, Kirchhoff-Love problem for cloth simulation). Even under this favourable setup, NeuRBF failed to converge: Poisson reconstruction plateaued at PSNR/SSIM $10.85/0.17$ versus $29.44/0.87$ for ours (Fig. I), and the Eikonal experiment did not recover a smooth surface (Fig. V). These observations reinforce that NeuRBF's reliance on hybrid architecture (Instant-NGP), signal-dependent initialisation and discontinuous interpolation make it unsuitable for PDE solving, whereas our differentiable grid is explicitly engineered for derivative-only supervision.

## B.2 COMPARISON WITH PINNS

We also compare against neural PDE solvers such as PINNs (Raissi et al., 2019). While PINNs perform reasonably on smooth signals, they struggle with high-frequency content, failing to capture wrinkles in cloth simulations or oscillations in Helmholtz. For instance, in the image reconstruction experiment, a GELU-based PINN achieves only PSNR/SSIM $\approx 19/0.65$, compared to $29.44/0.87$ with our method; results for image reconstruction from gradient field (Sec. 4.1) and Helmholtz equations (Sec. 4.2) are shown in Fig. IV. This limitation arises from their use of smooth activation functions (*e.g.*, Swish, GELU, tanh), which, though highly differentiable, are known to underfit high-frequency signals. We, therefore, adopt and primarily compare with Siren, which is better suited for such cases and widely used in physics-informed neural fields (Chen et al., 2023a; Kairanda et al., 2024).

## C EIKONAL EQUATION

In addition to the experiments presented in the main paper, we demonstrate the learning of 3D shapes represented as signed distance functions (SDFs) by solving the Eikonal equation.

An SDF defines a mapping from 3D space to the distance of a point from a watertight surface, with the zero level set corresponding to the surface itself. Rather than assuming access to ground-truth SDF values, we explore fitting SDFs directly to oriented point clouds. This task corresponds to solving the Eikonal boundary value problem, a non-linear PDE, that enforces the constraint $\|\nabla_{\mathbf{x}}\mathbf{u}(\mathbf{x})\| = 1$ almost everywhere in the domain. Following the training procedure described in Sitzmann et al. (2020), we formulate the loss function as:

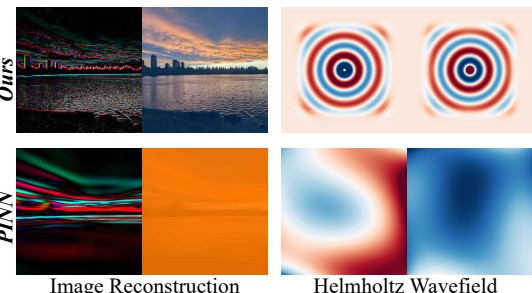

Image Reconstruction  Helmholtz Wavefield

**Figure IV: Comparison to PINN.** We compare $\partial^{\infty}$-*Grid* to MLP with GELU activation function, which struggles with the high-frequency content.

$$\mathcal{L}_{\text{eikonal}} = \int_{\Omega} |\|\nabla_{\mathbf{x}}\mathbf{u}(\mathbf{x})\| - 1| \, d\mathbf{x} + \int_{\Omega_0} \left(\|\mathbf{u}(\mathbf{x})\| + [1 - \langle\nabla_{\mathbf{x}}\mathbf{u}(\mathbf{x}), \mathbf{n}(\mathbf{x})\rangle]\right) d\mathbf{x} + \int_{\Omega\setminus\Omega_0} \psi(\mathbf{u}(\mathbf{x})) \, d\mathbf{x},$$

(13)

where $\Omega$ denotes the full domain, and $\Omega_0$ refers to the zero level set of the SDF. The regularisation function is defined as $\psi(\cdot) = \exp(-\alpha \cdot |\mathbf{u}(\cdot)|)$, with $\alpha \gg 1$, which penalises off-surface points that produce SDF values close to zero. As illustrated in Figs. II and V-(right), $\partial^{\infty}$-*Grid* is able to accurately solve the Eikonal equation and recover high-quality SDFs from oriented point clouds.

Although the encoder lives on a spatio-temporal grid $[-1, 1]^3$, we support irregular geometries by sampling PDE residuals and boundary terms on the target (potentially unstructured) domain $\Omega_0$. This works because $\mathbf{u}(\mathbf{x})$ in Eq. (5) is interpolated continuously, so arbitrary off-grid query sets (points and normals) can be supervised, enabling fits to surfaces (2D manifolds in 3D) from oriented point clouds. The trade-off is efficiency: empty grid points still store feature parameters, making our grid less parameter-efficient than coordinate-based MLPs.

### C.1 IMPLEMENTATION DETAILS

**Dataset.** In the SDF experiment, we learn a mapping $\mathbf{u}(\mathbf{x}) : [-1, 1]^3 \to \mathbb{R}$ using the loss function described above. The dataset $\mathcal{D} = \{(\mathbf{x}_i, \mathbf{n}(\mathbf{x}_i))\}_i$ comprises samples on the point cloud $\mathbf{x}$ and their ground-truth surface normals $\mathbf{n}(\mathbf{x})$.

**Architecture.** The feature encoder comprises $S = 4$ learnable grids, with the finest grid having a resolution of $N_{\max} = 56$, and each grid possessing a feature dimension of $F = 1$. Each query point is interpolated using $6^3 = 216$ features per scale. The feature decoder is a linear layer that maps the 3D input features to a 1D SDF.

**Training.** We employ stratified sampling over the domain, using 56 training samples along each input dimension. Additionally, an equal number of points are sampled on the surface, resulting in a total batch size of $2 \times 56^2$. A learning rate of $10^{-2}$ is used with the Adam optimiser. The SDF converges within 3000 iterations, requiring approximately six minutes of training time. For visualisation, the predicted SDF is evaluated on a uniformly sampled $112 \times 112$ grid covering the same domain, and meshes are extracted using the Marching Cubes algorithm.

### C.2 COMPARISON WITH GRID-BASED METHODS

Many grid-based methods (Müller et al., 2022; Li et al., 2023) represent signed distance functions (SDFs) using linear interpolation. In contrast to solving the Eikonal equation from oriented point clouds as in Eq. (13), these methods usually overfit to ground-truth SDFs computed from meshes, typically using a loss of the form:

$$\mathcal{L}_{\text{sdf}} = \int_{\Omega} \|\mathbf{u}(\mathbf{x}) - \text{SDF}(\mathbf{x})\| \, d\mathbf{x},$$

(14)

**Figure V: Solutions to the Eikonal equation from oriented point clouds.** For each method, we visualise 2D slices (left) and the mesh extracted from the SDF (right). Instant-NGP and NeuRBF can overfit when ground-truth SDF supervision ($\mathcal{L}_{\text{sdf}}$) is available, but they cannot reliably solve the Eikonal equation given partial point-cloud observations ($\mathcal{L}_{\text{eikonal}}$) because their gradients are not differentiable across grid boundaries (Fig. III). In contrast, our smooth formulation handles these gradients and solves the Eikonal problem comparably to baselines such as a sinusoidal MLP.

where the loss penalises deviations from the ground-truth SDF. This approach is often used either directly (Müller et al., 2022; Chen et al., 2023b) or indirectly, for example, in multi-view surface reconstruction (Li et al., 2023; Wang et al., 2023) (sometimes with additional regulariser enforcing the Eikonal condition). While such methods can produce accurate reconstructions when provided with high-quality ground-truth SDFs, they struggle when solving the Eikonal equation directly using $\mathcal{L}_{\text{eikonal}}$ (Eq. (13)). In particular, the Eikonal loss is not smoothly optimised across grid boundaries when using linear interpolation, which often results in incorrect SDF values (e.g., incorrect sign) on either side of the boundary. This issue is visualised in 2D slices for Instant-NGP in Fig. V. In contrast, our method employs differentiable grid interpolation, enabling the recovery of consistent and smooth SDFs directly from oriented point clouds.

## D    POISSON EQUATION

### D.1    IMPLEMENTATION DETAILS

**Dataset.**    We use the Camera image from sci-kit (Pedregosa et al., 2011). Following Siren (Sitzmann et al., 2020), the ground-truth gradient image is computed using the Sobel filter and scaled by a constant factor of 10 for training. The ground-truth Laplacian image is computed using a Laplace filter and scaled by a factor of $10k$.

**Architecture.**    The feature encoder comprises $S = 4$ learnable grids, with the finest grid of the resolution $N_{\max} = 512$, and each grid of the feature dimension $F = 2$. We vary these hyperparameters for gradient and Laplacian supervision; see Tab. 1 for the number of trainable parameters. Each query point is interpolated using $6^2 = 36$ features per scale. The feature decoder is a linear layer.

**Training.**    We train by evaluating on every pixel of the gradient or Laplacian image at each iteration. With regular fixed sampling over the domain, *i.e.*, 512 training samples along each input dimension, results in a total batch size of $512^2$. A learning rate of $10^{-3}$ is used with the Adam optimiser. The convergence varies per experiment and is visualised in Fig. 4 and Tab. 1.

## E    HELMHOLTZ EQUATION

We describe the full details of the Helmholtz experiment, as provided in Sec. 4.2 of the main paper. The Helmholtz equation is a second-order partial differential equation commonly used to model steady-state wave behaviour and diffusion phenomena. It is expressed as:

$$\left(\Delta + m(\mathbf{x})\,\omega^2\right)\mathbf{u}(\mathbf{x}) = -g(\mathbf{x}), \tag{15}$$

where $g(\mathbf{x})$ denotes a known source function, $\mathbf{u}(\mathbf{x})$ is the unknown complex-valued wavefield, and $m(\mathbf{x}) = \frac{1}{c(\mathbf{x})^2}$ represents the squared slowness, defined in terms of the wave velocity $c(\mathbf{x})$. We solve for the wavefield $\mathbf{u}$ using the $\partial^{\infty}$-*Grid* representation, with results presented in Figs. 1 and 5 and

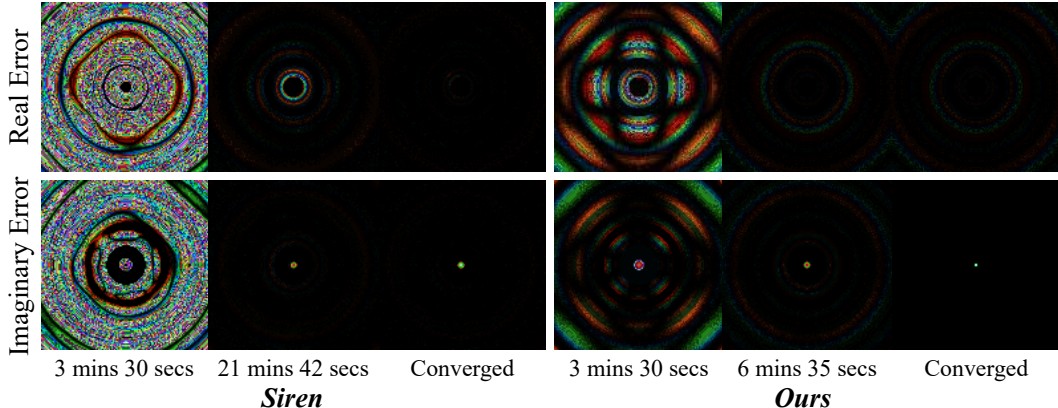

**Figure VI: $\ell_2$-error for the solution to the Helmholtz equation**. As presented in Fig. 5, while both ours and (Sitzmann et al., 2020) closely match the reference solution, our method trains significantly faster.

error visualisation in Fig. VI. In the following, we outline the specific formulation of the Helmholtz equation variant used in our implementation.

### E.1 PERFECTLY MATCHED LAYERS FORMULATION

To obtain a unique solution to the Helmholtz equation within a bounded domain, we adopt a perfectly matched layer (PML) strategy, following the formulation introduced in (Chen et al., 2013; Sitzmann et al., 2020). The PML serves to absorb outgoing waves at the domain boundaries, thereby preventing artificial reflections. This results in the following variant of the Helmholtz equation:

$$\frac{\partial}{\partial x_1}\left(e_{x_1}e_{x_2}\frac{\partial \mathbf{u}(\mathbf{x})}{\partial x_1}\right) + \frac{\partial}{\partial x_2}\left(e_{x_1}e_{x_2}\frac{\partial \mathbf{u}(\mathbf{x})}{\partial x_2}\right) + e_{x_1}e_{x_2}k^2\mathbf{u}(\mathbf{x}) = -g(\mathbf{x}), \tag{16}$$

where $\mathbf{x} = (x_1, x_2) \in \Omega$, the complex coefficients are given by $e_{x_i} = 1 - j\frac{\sigma_{x_i}}{\omega}$, and $k = \omega/c$. The damping term $\sigma_{x_i}$ along each spatial axis is specified as:

$$\sigma_{x_i} = \begin{cases} a_0\omega\left(\frac{l_{x_i}}{L_{\text{PML}}}\right)^2 & \text{if } x_i \in \partial\Omega, \\ 0 & \text{otherwise,} \end{cases} \tag{17}$$

where $a_0$ determines the attenuation strength (set to $a_0 = 5$), and $c = 1$ is the wave propagation speed. The term $l_{x_i}$ represents the distance to the nearest PML boundary along the $x_i$-axis, while $L_{\text{PML}}$ denotes the total PML width. The PML is applied exclusively near the domain boundary, defined as $\partial\Omega = \{\mathbf{x} \mid 0.5 < \|\mathbf{x}\|_\infty < 1\}$, and the original Helmholtz equation remains unchanged elsewhere. We optimise Equation 16 using the loss function introduced in the main text (Eq. 9-(main)), with the spatial weighting factor set to $\lambda(\mathbf{x}) = k = \frac{\text{batch size}}{5 \times 10^3}$.

### E.2 IMPLEMENTATION DETAILS

**Dataset.** In the Helmholtz experiment, we learn a mapping $\mathbf{u}(\mathbf{x}) : [-1, 1]^2 \to \mathbb{R}^2$ using the loss function described above. The dataset $\mathcal{D} = \{(\mathbf{x}_i, g(\mathbf{x}_i))\}_i$ comprises sampled spatial coordinates $\mathbf{x}_i$ and values of the Gaussian source function, defined as $g(\mathbf{x}) = \mathcal{N}(\mathbf{x}; [0, 0], 10^{-4})$.

**Architecture.** The feature encoder comprises $S = 4$ learnable grids, with the finest grid having a resolution of $N_{\max} = 128$, and each grid possessing a feature dimension of $F = 2$. Each query point is interpolated using $6^2 = 36$ features per scale. The feature decoder is an MLP that takes the concatenated features as input, projects them to a hidden layer of width 64 with Swish activation, and produces a 2D output through a final linear layer. The total number of trainable parameters is approximately $45.19k$.

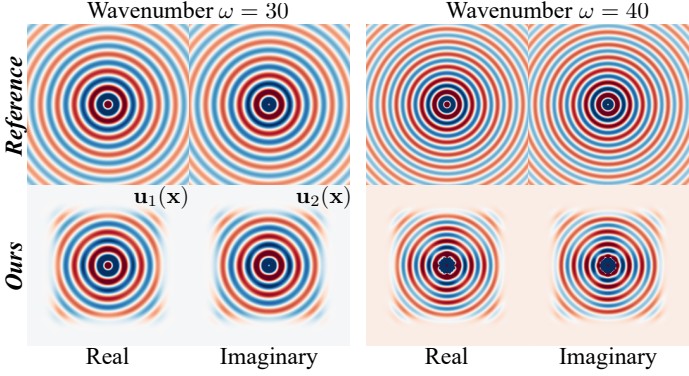

| $\omega$ | Real Error | Imag Error |
|---|---|---|
| 20 | 0.0008 | 0.0007 |
| 30 | 0.0040 | 0.0040 |
| 40 | 0.0250 | 0.0240 |

**Table I: Helmholtz equation with higher wavenumbers:** We report $\ell_1$ error with the reference solution for the real and imaginary parts of the wavefield. Parameter count: 45.19k for $\omega = 20, 30$; 176.71k for $\omega = 40$.

**Figure VII:** Results for Helmholtz equation with $\omega = 30, 40$ compared with closed-form solutions. Our $\partial^\infty$-*Grid* method remains accurate overall, with minor errors near the source at $\omega = 40$.

**Training.** We employ stratified sampling over the domain, with 256 training samples along each input dimension, resulting in a total batch size of $256^2$. A learning rate of $1 \times 10^{-3}$ is used with the Adam optimiser. The Helmholtz wavefields converge within 2500 iterations, requiring approximately 6 minutes of training time. For visualisation, the predicted wavefield is evaluated on a uniformly sampled $256 \times 256$ grid covering the same domain.

### E.3 Higher Wavenumber

To further evaluate our method under more challenging settings, we reproduce the Helmholtz equation experiments with higher wavenumbers, $\omega$. While Fig. 5 presented results for $\omega = 20$, here we extend the experiments to $\omega = 30$ and $\omega = 40$. Comparisons with the closed-form reference solutions are shown in Fig. VII and Tab. I. For $\omega = 30$, our method closely matches the analytical solution, whereas for $\omega = 40$ we observe noticeable errors near the wave source but otherwise maintain good accuracy across the domain.

## F Cloth Simulation

We describe the further details of the cloth simulation experiment, as presented in Section 4.3 of the main paper. Given the rest state of the cloth and the material elastic parameters $\mathbf{\Phi}$, quasistatic cloth simulation involves finding the equilibrium deformation field $\mathbf{u} : \Omega \to \mathbb{R}^3$ of the midsurface under the influence of external forces $\mathbf{g}$ and boundary constraints on $\partial\Omega$. The stable equilibrium configuration is obtained by minimising the total potential energy functional subject to Dirichlet boundary constraints:

$$\mathbf{u}^* = \arg\min_{\mathbf{u}} \int_\Omega \Psi[\boldsymbol{\varepsilon}(\mathbf{u}(\mathbf{x})), \boldsymbol{\kappa}(\mathbf{u}(\mathbf{x})); \mathbf{\Phi}] - \mathbf{g} \cdot \mathbf{u}(\mathbf{x}) \, d\mathbf{x},$$

$$\mathbf{u}(x_1, x_2) = \mathbf{b}(x_1, x_2) \quad \text{on } \partial\Omega,$$

$$(18)$$

where $\Psi$ is the internal elastic energy density, and $\boldsymbol{\varepsilon}, \boldsymbol{\kappa}$ represent the membrane and bending strain components, respectively. We solve for the deformation field $\mathbf{u}$ using the $\partial^\infty$-*Grid* representation. Results are shown in Figures 1 and 6 of the main paper.

### F.1 Implementation Details

**Dataset.** In the simulation experiment, we learn a mapping $\mathbf{u}(\mathbf{x}) : [-1, 1]^2 \to \mathbb{R}^3$ using the loss function described above. The dataset $\mathcal{D}$ consists of sampled spatial coordinates $\mathbf{x}$ and their corresponding rest state positions, set as $[x_1, x_2, 0]$. The material properties are defined by the parameter set $\mathbf{\Phi}$, with values:$\rho = 0.144, h = 0.0012, E = 5000$, and $\nu = 0.25$. We impose Dirichlet boundary conditions as hard constraints (following NeuralClothSim (Kairanda et al., 2024)) by displacing the top-left and top-right vertices of the cloth towards the centre of the input domain by 0.4 units each. The external force is modelled as a constant gravitational acceleration $[0, -9.8, 0]$.

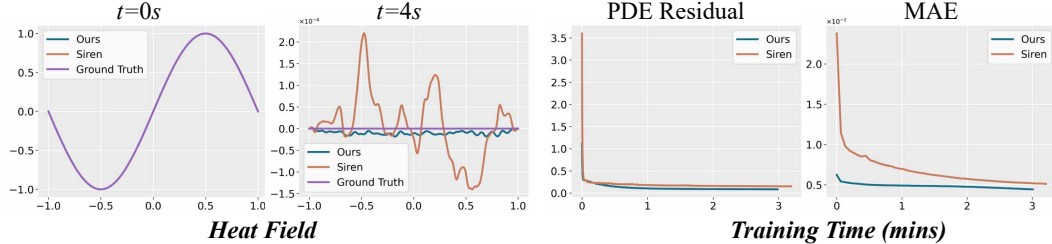

**Figure VIII: Solutions of the Heat Equation.** We solve the spatio-temporal heat equation successfully, where an initial sinusoidal wave diffuses over time (visualised at $t = 4s$). Compared to a coordinate-based MLP (Sitzmann et al., 2020), ours converges to the analytical solution more accurately and efficiently, as demonstrated with PDE residual and mean absolute error of the heat field.

Table II: **1D Advection** comparison with INSR (Chen et al., 2023a) and Grid solver (Fedkiw et al., 2001).

| Method | Error (MAE) | Training Time | Memory |
|---|---|---|---|
| INSR (Chen et al., 2023a) | 0.0030 | 5.33h | 3.53KB |
| Grid (Fedkiw et al., 2001) | 0.0029 | 1.80s | 27.35KB |
| Our $\partial^\infty$-*Grid* approach | 0.0023 | 1.5mins | 16.52KB |

**Architecture.** The feature encoder consists of a single learnable grid ($S = 1$) with a feature dimension of $F = 4$. We use a grid resolution of $N = 32$ for the simulation in Fig. 1-(main), and $N = 64$ for that in Fig. 6-(main). Each query point is interpolated using $6^2 = 36$ features per scale. The feature decoder is a linear layer that maps the 3-dimensional input features to a 3D deformation field. The total number of trainable parameters is approximately $4.37k$ for the results shown in Fig. 1-(main), and $16.91k$ for those in Fig. 6-(main).

**Training.** Similar to the Helmholtz experiment, we employ stratified sampling over the domain, using 64 and 128 training samples along each input dimension for Figs. 1 and 6-(main), respectively. A learning rate of $1 \times 10^{-2}$ is used with the Adam optimiser. The cloth simulation converges within 500 iterations, requiring at most 5 minutes of training time. For visualisation, the predicted deformation field is evaluated on a uniformly sampled $64 \times 64$ grid spanning the same domain.

## G HEAT EQUATION

Here, we additionally demonstrate our result for the 1D heat equation. The heat equation is a diffusion PDE used to model temporal smoothing of signals:

$$\frac{\partial u}{\partial t} = \alpha \frac{\partial^2 u}{\partial x^2}, \tag{19}$$

where $u(x, t)$ represents the temperature at position $x$ and time $t$, and $\alpha$ is the thermal diffusivity constant. We consider a spatial domain $x \in [-1, 1]$ and a temporal domain $t \in [0, 4]$ with Dirichlet boundary conditions and an initial sinusoidal condition $u(x, 0) = \sin(\pi x)$ with $\alpha = 1$. It admits an analytical solution:

$$u(x, t) = e^{-\alpha \pi^2 t} \sin(\pi x), \tag{20}$$

which we use to quantitatively evaluate reconstruction fidelity. We train with the PDE loss with hard boundary constraints. Fig. VIII summarises the experiment: the sinusoidal wave diffuses as expected and matches the analytical solution with a mean absolute error of 0.004 over uniformly sampled space-time points. Relative to the coordinate-based MLP baseline (Sitzmann et al., 2020), the PDE residual and the heat-field error reduce faster. Thus, $\partial^\infty$-*Grid* faithfully recovers the spatial and temporal evolution of the diffusion process.

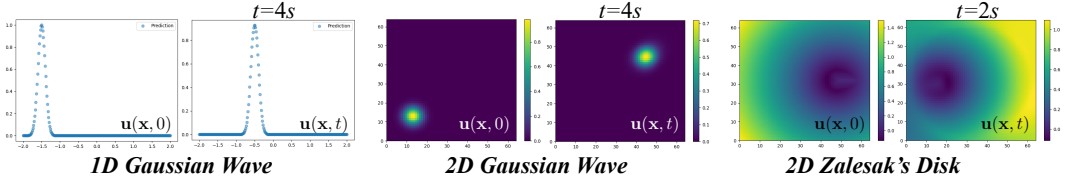

**Figure IX:** We solve spatio-temporal advection successfully for various 1D and 2D fields, including 1D and 2D Gaussian waves as well as 2D Zalesak's disk.

## H    ADVECTION EQUATION

We consider the spatio-temporal advection equation

$$\frac{\partial \mathbf{u}}{\partial t} + \mathbf{a} \cdot \nabla_{\mathbf{x}} \mathbf{u} = 0, \tag{21}$$

which describes the transport of a field with advection velocity $\mathbf{a}$. We demonstrate our method on 1D and 2D cases, including Gaussian wave propagation and the Zalesak benchmark with a challenging discontinuous boundary.

### H.1    1D ADVECTION

We follow the INSR setup (Chen et al., 2023a), where a Gaussian-shaped wave, centred at $\mu = -1.5$ propagates rightwards in the spatial domain $[-2, 2]$, with advection velocity $a = 0.25$. In contrast to INSR, we impose boundary and initial conditions as hard constraints and treat time as a continuous input to our feature grid, rather than discretising it (they use midpoint integration). We present the results in Tab. II and Fig. IX. Our results closely match both the analytical solution and INSR, with a mean absolute error (MAE) reported at $t = 4$ seconds. The INSR and the finite difference-based grid solver (Fedkiw et al., 2001) values are taken from Tab. 1 of (Chen et al., 2023a). $\partial^{\infty}$-*Grid* suffers minimal dissipation; we observe an MAE of $2.3 \times 10^{-3}$ at $t = 4$s, *i.e.*, less than $0.3\%$ amplitude loss compared to the analytical solution, while still training $200\times$ faster than INSR. Overall, our approach achieves lower error relative to the ground truth, reduced training time compared to INSR, and lower memory use compared to numerical solvers.

### H.2    2D ADVECTION

For the 2D Gaussian wave, we adopt the domain and initial conditions of Chetan et al. (2025), simulating for 4 seconds. Relative to the closed-form solution, our method achieves mean advection errors of 0.0047 (after 3 minutes), 0.0032 (after 6 minutes), and 0.00246 (after 15 minutes, fully converged); see Fig. IX-(centre) for qualitative results. Note that in Chetan et al. (2025) advection example, they employ forward/explicit Euler (possible with a post hoc gradient operator) for timestepping, unlike INSR (Chen et al., 2023a) (which can support mid-point due to underlying Siren representation) and ours (which supports auto diff and mid-point similar to Siren/INSR).

### H.3    ZALESAK'S DISK PROBLEM

For the Zalesak benchmark, we follow the setup in Dupont & Liu (2003), and solve for the signed distance function (SDF) of the slotted disk under advection. The results converge and preserve the overall shape, although some numerical diffusion is visible as blurring along the slot edges; see Fig. IX-(right). After half a revolution, we obtain an $\ell_1$-error of 0.065 compared to the closed-form rigid-body rotation solution with constant angular velocity.

## I    LIMITATIONS AND FUTURE WORK

While our method leverages differentiable interpolation to enable accurate representation of signals and their derivatives, it exhibits several limitations. First, the interpolated features are sensitive to the choice of the Gaussian RBF shape parameter, which in turn determines the effective neighbourhood.

As a result, the representation can introduce some degree of smoothing, depending on this configuration. While we observed convergence across all experiments, further investigation is needed to establish theoretical guarantees for $C^\infty$ under neighborhood truncation. A well-studied drawback of RBF methods, particularly near the domain boundaries, is the occurrence of boundary errors (Runge's phenomenon), which can negatively impact reconstruction accuracy in those regions. Our grid approach also suffers from the curse of dimensionality; for example, fitting SDFs in 3D is notably slower than in 2D. The efficiency gain of our method is primarily due to the use of structured Cartesian grids, which is lost when solving PDEs on manifolds. As a direction for future work, we suggest exploring planar projection strategies to improve scalability and computational efficiency in high-dimensional domains.

Moreover, our novel representation, although it improves speed over neural solvers and introduces differentiability into feature grids, it might not be as efficient as traditional numerical solvers, such as multi-grid methods. Since this requires further thorough investigation, in our evaluations, we provided initial experiments and comparisons to numerical methods help guide future work in this direction. In the future, a custom CUDA kernel implementation of $\partial^\infty$-*Grid* could further accelerate optimisation compared to our current PyTorch implementation.

