# OpenReview forum: "$\boldsymbol{\partial^\infty}$-Grid: A Neural Differential Equation Solver with Differentiable Feature Grids"
_ICLR.cc/2026/Conference — ICLR 2026 Poster_

### Official Review · Reviewer_mShH · 2025-10-21

**Soundness:** 2
**Presentation:** 3
**Contribution:** 3
**Rating:** 6
**Confidence:** 4

**Summary:**

The submission introduces ∂∞-Grid, a differentiable grid-based neural representation for solving partial differential equations (PDEs).Unlike coordinate-based multilayer perceptrons (MLPs) such as PINNs, ∂∞-Grid employs Radial Basis Function (RBF) interpolation over learned feature grids, resulting in an infinitely differentiable representation that supports analytic computation of spatial derivatives.
The framework further integrates multi-resolution, co-located grids to facilitate gradient propagation across scales.

Empirical evaluations encompass a range of PDE problems including gradient-based reconstruction, Helmholtz, Kirchhoff–Love plate deformation, Eikonal, and advection equations and demonstrate 5–20× faster convergence than coordinate-based baselines, with comparable or improved accuracy.
The approach aims to unify the strengths of grid-based representations, efficient yet non-differentiable, and coordinate-based networks, which are flexible but computationally slower.

**Strengths:**

1.	The combination of feature grids with Gaussian RBF interpolation is intersting, technically sound and conceptually straightforward. Since Gaussian RBFs are $C^{\infty}$, the method inherently supports analytic higher-order derivative computation, which is important for PDE solvers.
2.	The experiments report significant improvements in convergence rate and accuracy, achieving 5–20x faster training compared to SIREN and K-Planes. These speedups represent a substantial computational gain, reducing training time from hours to minutes or even seconds, and demonstrate the method’s scalability and efficiency across diverse PDE formulations.
3.	The presentation is clear, and the methodology is described in a structured and coherent manner.
4.	The experimental evaluation across multiple classes of PDEs suggests that the proposed approach is broadly applicable rather than tailored to a specific problem type.

**Weaknesses:**

1.	The formulation assumes a Cartesian grid. In contrast to coordinate-based PINNs, ∂∞-Grid cannot directly address irregular geometries or complex boundary conditions without additional coordinate transformations or masking procedures, limiting its applicability to structured domains.
2.	The comparisons focus mainly on SIREN and K-Planes. Incorporating recent operator-learning approaches such as the Fourier Neural Operator or DeepONet would strengthen the evaluation, as the proposed framework requires solving each new boundary condition or PDE configuration from scratch, unlike operator-based methods that generalize across parameter variations.
3.	The comparison to K-Planes may not be entirely fair for derivative-based tasks, as K-Planes relies on non-differentiable piecewise linear interpolation.
4.	The paper lacks an analysis of how the solver’s performance varies with grid resolution, the number of multi-resolution levels, and the dimensionality of the feature vectors. It is also unclear how these parameters are selected.
5.	While the paper presents interesting empirical results, a more formal treatment of aspects such as convergence, stability, or error bounds relative to classical numerical solvers or PINNs could strengthen the contribution. Clarifying the underlying functional space (e.g., RKHS, $C^2$) would also enhance the theoretical grounding.

Minor comments

1.	The symbol f is used for different quantities in Eqs. (1) and (2). Although boldface differentiates the latter, this overlap could cause ambiguity.
2.	The “Poisson” experiment corresponds to a gradient or Laplacian-based reconstruction task rather than a direct PDE solution; clearer terminology would help prevent confusion.

**Questions:**

1.	Can ∂∞-Grid be extended to non-rectangular or manifold domains?
2.	Could the framework be combined with coordinate-based PINNs to handle complex geometries while preserving grid efficiency?
3.	Could the authors clarify the implementation details for both K-Planes and Instant-NGP? For example, whether K-Planes is formulated as
$u(x) = \mathrm{MLP}(\mathrm{linear_interpolation}(F, x))$.

5.	Could the authors evaluate or discuss the sensitivity of the proposed approach to grid resolution, the number of scales, and the dimensionality of the feature vectors?

6.	Figure 5: Could the authors include an image illustrating the difference between SIREN and the proposed method?
7.	What specific classes of PDEs can the method address effectively, and under what assumptions regarding smoothness or boundary conditions?

---

> ### Author Response · Authors · 2025-11-28
> **Rebuttal mShH (1/2)**
>
> ### Complex or Non-rectangular Geometries
> Although the encoder is defined on a spatio-temporal Cartesian grid $[-1,1]^d$, we already support irregular geometries by restricting the PDE residual and boundary samples to the desired potentially unstructured input domain. This works because $\mathbf{u}(\mathbf{x})$ is interpolated continuously via RBFs before decoding (Eq. 2), so arbitrary query sets can be supervised. As a concrete example, the Appendix C Eikonal experiment fits watertight surfaces (2D manifolds embedded in 3D) from oriented point clouds (where SDF and normals are enforced at irregular points in the input domain). Moreover, our boundary formulations (Eq. 5) also support arbitrary points/curves/surfaces, showing that embedded manifolds and unstructured domains behave similarly to coordinate-based PINNs. The outstanding limitation is efficiency, not feasibility: empty grid points still consume feature parameters. We now highlight this trade-off (Appendix C). Note that coupling the grid features with learned coordinate mappings (e.g., hash grids or TensORF-style tensor decompositions) is a natural next step for scaling to higher-dimensional fields or more curved manifolds, as already discussed in our Conclusions/Limitations sections.
>
> ### Comparisons to Operator-learning Baselines
> We thank the reviewer for asking about FNO/DeepONet. Our method is a neural solver, not a neural operator [Hao et al.]: the goal of our new representation is to optimise a single PDE instance directly from its residual without requiring a dataset or paired solutions. In contrast, the architectural goal of operator-learning approaches is generalisation across parameter variations rather than per-instance solving accuracy. Because of this mismatch, a side-by-side comparison would be unfair—operator learners assume access to ground-truth solutions during training (e.g., DeepONet), whereas our setting only provides residuals. That said, the two directions are complementary: a learned operator could provide an initial guess that our differentiable grid could refine under physics supervision.
>
> Hao, Zhongkai, et al. "Physics-informed machine learning: A survey on problems, methods and applications." arXiv preprint arXiv:2211.08064 (2022).
>
> ### K-Planes Comparison
> We agree that K-Planes (and Instant-NGP) are not designed for derivative-based supervision, but they remain the most relevant grid baselines and directly motivate our differentiable interpolation. We employ the exact architecture proposed in \citep{keil2023kplanes}, i.e., $u(\mathbf{x}) = \mathrm{MLP}(\mathrm{linear\_interpolation}(F,\mathbf{x}))$, and optimised it with the same PDE losses as ours. Appendix A now documents the exact hyperparameters for both K-Planes and Instant-NGP so that the derivative-based comparisons are reproducible and transparent.
>
> ### Sensitivity to Scales
> Section 4.5 and Fig. II already analyse the effect of multi-resolution grids: adding scales accelerates global gradient propagation and stabilises convergence, yielding a $6\times$ faster reconstruction for the gradient-supervised image task and smoother Eikonal solves. We also report how varying the RBF neighbourhood radius $ \rho $ and shape parameter $ \varepsilon $ impacts accuracy/time in the ablations of Sec. 4.5 (Table 2).

---

> ### Author Response · Authors · 2025-11-28
> **Rebuttal mShH (2/2)**
>
> ### Convergence, Stability, and Function Space Assumptions
> The draft already demonstrates a broad set of PDE classes: elliptic (Poisson, Helmholtz), parabolic (heat), hyperbolic (advection), and both linear (Helmholtz, advection–diffusion) and highly non-linear problems (Eikonal, Kirchhoff–Love). Our Gaussian basis is $C^\infty$, which is sufficient for the higher-order PDEs we solve, and we cover both strong-form residuals (e.g., Poisson) and weak-form losses (cloth simulation). Residual errors appear directly in the loss, and Dirichlet boundaries are enforced through the hard constraint term, so one loss template suffices across experiments, which integrates initial/boundary conditions.
>
> We recognise that classical numerical solvers provide a deeper theoretical guarantee, whereas neural PDE solvers remain an emerging area, i.e., physics-informed machine learning (PIML), for example, still face open questions on convergence and stability (see recent surveys such as [Karniadakis et al.]). Our focus is on computational efficiency, which is one of the open problems in PIML and complementary to those theoretical efforts: advances in neural solver regularisation, variational formulations, or loss design can be incorporated into our framework because the formulations are orthogonal. A more formal analysis of convergence/stability for our differentiable grid is an important direction, and we will flag this as future work while citing representative theory papers (e.g., NTK analyses, [Jacot et al.]).
>
> [Jacot et al.] A. Jacot, F. Gabriel, and C. Hongler, “Neural tangent kernel: Convergence and generalization in neural networks,” Advances in neural information processing systems, vol. 31, 2018.
> [Karniadakis et al.] Karniadakis, George Em, et al. "Physics-informed machine learning." Nature Reviews Physics 3.6 (2021).
>
> ### Symbol f
> Thanks for catching this clash. We now denote the known function as $g$ in Eq. (1), the method section, and all subsequent experimental descriptions, while retaining $\mathbf{f}$ exclusively for the feature encoder.
>
> ### Poisson experiment
> Our experiment of reconstructing an image from its Laplacian does correspond to solving the Poisson PDE, which is one of the standard applications of the Poisson equation in graphics (other e.g., Poisson surface reconstruction). We agree that the gradient-only experiment does not fully belong in this class, and instead, we consider it as a simpler reconstruction variant to illustrate the same intuition. Accordingly, we rewrote Sec. 4.1 (Fig. 4, I, IV; Table 2) to use consistent terminology, explicitly stating that the loss evaluates the Laplacian residual, and clarifying that the gradient-only variant is analysed additionally. We hope this wording makes it clear that our method still enforces a Poisson residual whenever the Laplacian is used.
>
> ### Figure 5: Visualising Siren vs. ∂∞-Grid
> We now explicitly illustrate the difference between our method and Siren for the Helmholtz equation in Fig. VI. The figure reports $\ell_2$ error maps against the reference solution for both Siren and our approach mid-training and after convergence. Compared to Siren, we observe slightly higher error on the real part but lower error on the imaginary part, and, crucially, our method still trains significantly faster.

---

### Official Review · Reviewer_qdmK · 2025-10-28

**Soundness:** 4
**Presentation:** 4
**Contribution:** 2
**Rating:** 8
**Confidence:** 4

**Summary:**

The paper proposes a differentiable grid-based representation for solving differential equations. By combining multi-resolution feature grids with smooth radial basis function (RBF) interpolation, the method enables higher-order differentiability, allowing it to learn continuous fields whose derivatives satisfy physical constraints. The approach is applied to a variety of PDE problems, including the Poisson equation, the Helmholtz equation, and the Kirchhoff–Love model, achieving faster convergence than coordinate-based neural solvers such as SIREN.

**Strengths:**

- The paper is written with clarity, organization, and completeness. Overall, the derivation is straightforward and easy to follow.

- The paper includes a comprehensive and thorough list of experiments, including Poisson Equation, Helmholtz Equation, Cloth Simulation, Eikonal Equation, and Advection and Heat Equations, and with both qualitative and quantitative results.

- I appreciated the numerous experiments the authors supplement in the Appendix, especially the section on Eikonal Equations.

- The method achieves comparable or better accuracy than MLP-based PINNs and SIREN while providing 5–20× faster training and inference.

**Weaknesses:**

- The use of RBF interpolation in feature grids is conceptually similar to prior work such as NeuRBF (Chen et al., ICCV 2023). Both frameworks ultimately learn continuous fields under differentiable consistency constraints, and while the authors claim NeuRBF "adapts to known target signals," this is not the case for NeRF-style trainings where the supervision comes from images and the ground truth 3D field is also unknown. I would sincerely suggest that the authors acknowledge this part and limit their contribution to "applying existing RBF feature grid to PDEs."

This is my main assessment of the concerns of the paper, but from a "solving PDEs" perspective, there is still merit in having this paper published.

**Questions:**

- How does the accuracy or convergence rate scale? Have the authors attempted any PDEs that require actual 3D fields (as opposed to the 2D-to-3D mapping in the cloth simulation)?

- Can the method handle PDEs with non-smooth solutions or discontinuous coefficients, where RBF smoothness may become a drawback?

- In the Eikonal equation experiment in the Appendix, how is the proposed method now different from NeuRBF? Would the authors be able to supplement a comparison with NeuRBF regarding this specific setup?

---

> ### Author Response · Authors · 2025-11-28
> **## Rebuttal qdmK**
>
> ### NeuRBF comparison
> Kindly see the common rebuttal.
>
> ### Scaling Accuracy to Full 3D PDEs
> Ours already handles full 3D fields, not just 2D$\rightarrow$3D mappings. Appendix C (Fig. 7, V) shows the 3D Eikonal solution from oriented point clouds. Appendix H.1, H.2 includes 2D advection experiments (Gaussian wave, Zalesak’s disk) where the domain $(x,y,t)$ is three-dimensional. The current prototype precomputes neighbourhoods for differentiable interpolation (with $\rho^d$ neighbours per scale), so scaling to higher 3D resolutions is memory-intensive. This is unlike non-differentiable linear interpolation, which avoids this overhead. We have acknowledged this in the limitations and note in the paper that future work will explore grid-scaling techniques such as hash grids or tensor decompositions for PDE supervision, analogous to advances in grid-based signal fitting.
>
> ### Robustness on Non-smooth PDEs
> Although Gaussian RBFs are smooth, we did not observe issues when solving non-smooth solution fields in the benchmarks we tested (e.g., image reconstruction with high-frequency content). Multi-resolution grids let us allocate higher spatial resolution where needed, so the representation can capture piecewise-smooth solutions. For the Helmholtz equation, the MLP decoder helps model the singularity at the centre of the wavefield, whereas the remaining experiments work well with a linear decoder. For the Eikonal SDF reconstruction, we do see some surface smoothing due to the coarse $56^3$ grid rather than the interpolation itself (Fig. V). Scaling to higher dimensions is challenging with RBFs because larger neighbourhoods must be interpolated, but smoothness itself is not an issue thanks to the multi-resolution hierarchy.

---

### Official Review · Reviewer_ai8J · 2025-10-30

**Soundness:** 3
**Presentation:** 3
**Contribution:** 3
**Rating:** 6
**Confidence:** 3

**Summary:**

The paper proposes a fast neural PDE solver based on grid-based neural fields like Instant NGP. Traditional grid-based neural fields are neural fields with features on a grid that are modulated by a small MLP. For arbitrary query points, these fields compute the feature by using d-linear interpolation. However, relying on d-linear interpolation means that these networks do not yield meaningful high-order derivatives. The authors propose to mitigate this by proposing a differentiable RBF-based interpolation that can yield accurate higher-order derivatives. The authors use a Gaussian RBF kernel to compute interpolation weights for the grid points surrounding a query point.

The authors apply their method to solving a variety of PDEs, ranging from the Helmholtz equation, the Advection and Heat Equation, Eikonal Equation, etc, showing increased accuracy over contemporary grid-based neural field architectures like Instant NGP and increased efficiency as compared to MLP-based architectures like SIREN.

**Strengths:**

1. The problem of obtaining accurate higher-order derivatives from grid-based neural fields is an important one, and the authors' differentiable interpolation scheme is a great step in this direction.
2. The authors have done a rigorous evaluation across a broad range of PDEs.

**Weaknesses:**

Weaknesses

1. Writing can be improved:
	- Some of the authors' claims about their contributions seem a bit exaggerated. For instance, multi-resolution grids were an idea that was proposed in both Instant NGP and K-planes. While the authors have applied this idea to a novel setting of PDE-solvers, claiming the idea itself as a contribution seems misleading to me.
	- Notational issue: it seems that $r$ is both a parameter (input to the RBF kernel) and a hyper-parameter (extent of neighborhood considered for kernel) - this should be either resolved or the explanation of the $r$ hyper-parameter should be improved.
	- For 2D Advection, the authors claim that their approach shows no numerical dissipation. However, the change in the peak value on the colorbar between the initial value and the final state shows that there has been some dissipation. The authors have also not shown quantitative results for 1D and 2D advection
2. Missing Results?: Could not find results for the heat equation in the supplementary material. (Appendix G)

**Questions:**

1. Appreciate the comparison to NeuRBF, but could the authors shed some light on how NeuRBF's formulation differs from their approach, which makes it possible for their approach to be used as an effective neural solver? The current explanation is unclear.
	- As per the authors' explanation it seems that NeuRBF's design makes it suitable to fit zeroth-order signals but not gradient-/derivative-based residuals like the losses described by the authors in their experiments which makes it difficult to use NeuRBF as a neural solver, but I am still not convinced by the author's explanation of on what specific design detail of NeuRBF makes it infeasible to use it as a neural solver.
	- Currently, the authors point to the requirement of ground-truth signals by NeuRBF as an issue, but that is because NeuRBF has been floated as a signal representation. Even if the authors' proposed architecture were to be used as a signal representation, for instance, to learn the initial conditions of a PDE, they would require the ground-truth signal for training.
2. Can the author's architecture be used for overfitting to any input signal as well? This could increase the scope of their contribution.
	- Asking because Instant NGP reported that moving to higher-order interpolation approaches hurt performance on reconstruction
	- If the authors' proposed approach can do better would be a great contribution to the community. Even if not, it would be something useful for the community to know.

I am generally leaning towards accepting this work. If the authors can address the weaknesses and provide an explanation for the above questions, I think that would make this work worthy of being accepted.

---

> ### Author Response · Authors · 2025-11-28
> **Rebuttal ai8J**
>
> ### NeuRBF comparison
> Kindly see the common rebuttal.
>
> ### Disambiguating the RBF Radius $r$
> Thank you for pointing out the ambiguity. In the revision, we now reserve $r$ for the radial argument of the Gaussian basis $\varphi(r)=\exp(-(\varepsilon r)^2)$, and denote the discrete neighbourhood ring by the new hyperparameter $\rho$ so that $\mathcal{N}_\rho(\mathbf{x})$ is unambiguously the set of grid nodes included in the interpolation. Table 2 reports performance for different $(\varepsilon,\rho)$ pairs, and Sec. 4.5 clarifies that our default choice is $\varepsilon=1,\rho=2$ or $3$.
>
> ### Quantifying Dissipation in Advection
> We softened the language in Appendix H.1 to “minimal dissipation.” Quantitative evidence is already in Table II, which compares our 1D advection benchmark against INSR and the grid solver: we obtain an MAE of $2.3\times10^{-3}$ at $t=4$ s, corresponding to under $0.3\%$ amplitude loss relative to the analytical solution while training $200\times$ faster than INSR. Appendix H.2 further reports mean advection errors of $4.7\times10^{-3}$, $3.2\times10^{-3}$, and $2.46\times10^{-3}$ for the 2D Gaussian case after 3, 6, and 15 minutes of optimisation, respectively. Finally, Appendix H.3 lists an $\ell_1$ error of $0.065$ for the Zalesak variant of the 2D advection problem against its closed-form solution; also see Fig. IX.
>
> ### Locating the Heat Equation Results
> Thanks for this suggestion. While we had mentioned obtaining an MAE of $4\times10^{-3}$ over $[-1,1]\times[0,4]$ against the closed-form solution for the 1D Heat PDE with sinusoidal initial conditions, we had indeed missed adding the full version of the results. Appendix G now contains the complete experiment: we solve on $[-1,1]\times[0,4]$ with hard boundary constraints and report the same MAE over uniformly sampled space-time points. Fig. VIII shows both the initial state and the $t=4$ snapshot alongside the PDE residuals for ours and the compared method (Siren).
>
> ### Credit for Multi-resolution Grids
> We do not position multi-resolution grids as a standalone contribution; Sec. 3.3 cites Instant-NGP/K-Planes as the source of this design. Our technical section merely states “Multi-resolution grids for faster global gradient flow” to explain why we reuse this idea in the PDE setting. Whereas prior grid methods emphasise parameter efficiency, our focus is on stabilising gradient propagation through differentiable RBF interpolation (without adaptive RBFs or per-node scale optimisation). The revision now explicitly attributes the grid hierarchy to prior work while clarifying that our novelty lies in coupling it with higher-order differentiable interpolation for PDE supervision.
>
> ### Using ∂∞-Grid for Signal Fitting
> Thank you for suggesting that we highlight this capability more clearly. The original submission had shown image fitting examples in the appendix; we have now expanded this into Sec. 4.4 of the main paper with Fig. 7 showcasing a colour image and SDF fitting (direct and gradient supervision). When ground-truth signals are available, we simply drop the PDE residual term and optimise reconstruction losses, so the same architecture handles both direct supervision and PDE supervision, such as Eikonal fits from oriented point clouds. Note that the neighbourhood ring $ \mathcal{N}_\rho $ can be reduced for pure signal fitting to trade accuracy for speed. A direct comparison with Instant-NGP could be inconclusive because their CUDA hash-grid implementation could lead to a performance difference from our PyTorch grid without hash tables; nevertheless, our qualitative results demonstrate that higher-order interpolation does not impede signal reconstruction.

---

### Author Response · Authors · 2025-11-28
**Common Rebuttal**

We sincerely thank the reviewing committee for the thoughtful reviews and constructive suggestions. We are encouraged by the positive assessments of the work’s significance and presentation, and we have refined the draft to incorporate the reviews.

The revised draft highlights all reviewer-specific changes in colour (ai8J in red, qdmK in green, mShH in blue, common ones in purple) to make the updates easy to track. Below, we summarise the main additions in terms of sections, visualisations, and experiments.

### Added Sections, Visualisations & Experiments (ai8J, qdmK, mShH)
1. **Signal Fitting (ai8J)** : Sec. 4.4, Fig. 7
2. **NeuRBF comparison for Eikonal Equation (qdmK):** Sec. 4.5, Appendix B, Figs. I, V
3. **Heat Equation results (ai8J):** Fig. VIII
4. **Siren vs. ∂∞-Grid difference for Helmholtz (mShH)**: Fig. VI
5. **Details on K-Planes/Instant-NGP Comparison (mShH)**: Appendix A

### NeuRBF comparison (ai8J, qdmK)
Thank you for requesting further details on the NeuRBF comparison. We have substantially expanded the discussion in App. B by illustrating the formulation difference in Fig. III and qualitative comparisons in Figs. I and V (all highlighted in purple); the key points are:

1. **Different objectives and RBF initialisation.** NeuRBF fits signals by relying on known targets (images/SDFs) or distilled proxies such as K-Planes/TensoRF (NeRF) to initialise the positions and shapes of its adaptive RBFs. Our solver instead uses fixed-shape Gaussian RBFs on grid nodes and therefore works purely from derivative supervision across PDEs, never seeing the ground-truth field (Sec. B; Figs. I, III, V).
2. **Interpolation behaviour.** NeuRBF restricts each query to a single adaptive RBF ring, so the interpolation becomes discontinuous whenever the neighbourhood changes (Fig. III); they additionally rely on Instant-NGP to maintain $C^0$ continuity. Extending their scheme to larger neighbourhoods is non-trivial. We precompute overlapping $ \mathcal{N}_{\rho} $ neighbourhoods on the grid, yielding $C^\infty$ interpolation suitable for PDE losses.
3. **Empirical evidence.** We adapted the official NeuRBF code by swapping in our gradient-based Poisson loss and the Eikonal loss while retaining their RBF initialisation. NeuRBF saturates at PSNR/SSIM $10.85/0.17$ versus $29.44/0.87$ for ours on Poisson, and fails to recover a smooth SDF from oriented point clouds (Figs. I, V). These failures persist even with favourable initialisation (which is possible for the gradient image and SDF tasks but not necessarily for other PDEs) because of the aforementioned interpolation discontinuities.

---

### Meta-Review · Area_Chair_vLry · 2025-12-22

**Summary:**

The paper presents a grid-based neural field using Gaussian RBF interpolation with a truncated, precomputed neighbourhood scheme, and reports encouraging empirical results on PDE-driven tasks. Reviewers generally agree the approach is practically appealing and empirically strong in several PDE settings, with reported speedups over coordinate-based MLP solvers such as SIREN and over grid baselines that use piecewise-linear interpolation.  Reviewer qdmK in particular finds the paper clear and the experimental coverage broad, while also noting conceptual similarity to NeuRBF and suggesting the paper should be positioned more as “applying an existing RBF feature-grid idea to PDE supervision.”  In rebuttal, the authors made a substantial effort to address missing experiments and clarity issues: they added signal-fitting results, expanded and formalized the NeuRBF comparison (including an Eikonal setup), added heat equation results, clarified ambiguous notation (separating the Gaussian shape parameter from the neighborhood hyperparameter), and softened over-claims (e.g., about dissipation)..  Despite these improvements, a key technical concern both by reviewers and myself is that the paper’s central “$C^\infty$/higher-order smoothness” framing is not aligned with what is actually guaranteed under the proposed truncated neighbor-set implementation, and that the efficiency narrative is heavily tied to structured Cartesian grids.

Overall, the work is useful and well executed as a practical neural representation for PDEs. The main remaining concerns are about rigor and framing, especially regarding what smoothness can actually be guaranteed under the proposed truncation scheme, and the paper’s precise positioning and assumptions (e.g., reliance on structured grids, the truncation regime, derivative order, and the relationship to NeuRBF and other kernel-based approaches). These issues are not purely editorial, but they appear addressable through careful clarification and revision. I recommend acceptance provided the final version tightens these points.

**Reviewer Concerns:**

1. The central “$C^\infty$/higher-order smooth” framing is problematic: the proposed efficiency comes from truncating a globally supported Gaussian kernel (i.e., a practical kernel-basis truncation). With neighbour-set truncation, the active set can change across grid/cell boundaries, so continuity is not guaranteed, let alone smoothness. This gap is not discussed adequately, yet the paper’s language suggests positive. Reviewers ai8J and qdmK raised related concerns and explicitly compared the approach to NeuRBF.

2. The precomputed neighbourhood indices and fixed-size “ring” lookup are primarily advantageous for Cartesian/structured grids, where neighbourhoods can be enumerated with regular offsets and reused efficiently. For unstructured meshes or point clouds, this benefit largely disappears: neighbour queries typically require kNN/radius search or variable-length indexing, undermining the efficiency narrative. The manuscript does not clearly delimit this dependence on structured discretizations.

3. The experiments suggest the method can work well in certain regimes, but the paper needs to re-center its contribution: clearly state when it is better than existing kernel/grid approaches (including NeuRBF), and under what assumptions (grid type, truncation regime, derivative order). As written, the positioning is not sufficiently precise.

**Reviewer Scores:**

I think three reviewers will likely keep the same scores.

---

### Decision · Program_Chairs · 2026-01-26

Accept (Poster)